

# Calibrating a New Attenuation Curve for the Dead Sea Region Using Surface Wave Dispersion Surveys in Sites Damaged by the 1927 Jericho Earthquake

Darvasi Yaniv[1], Agnon Amotz[1]

5   [1]The Fredy & Nadine Herrmann Institute of Earth Sciences, The Hebrew University of Jerusalem, 9190401, Israel

*Correspondence to*: Darvasi Yaniv (yaniv.darvasi@mail.huji.ac.il)

**Abstract.** Strong motion data is not common around the Dead Sea region. Therefore, calibrating a new attenuation equation is a considerable challenge. However, the Holy Land has a remarkable historical archive, attesting to numerous regional and local earthquakes. Combining the historical record with modern measurements will enhance the regional equation.

10   On 11 July 1927, a crustal rupture generated a moderate 6.25$M_L$ earthquake around the northern part of the Dead Sea. Up to five hundred people were killed and extensive destruction was recorded, even at places as far as 150 kilometers from the focus. We consider local near-surface properties, in particular, the shear-wave velocity, as an amplification factor. Where the shear-wave velocity is low, the seismic intensity at places far from the focus would likely be greater than expected from a standard attenuation curve. In this work, we used the Multi Analysis of Surface Waves (MASW) method to estimate seismic 15   wave velocity at anomalous sites in Israel in order to calibrate a new attenuation equation for the Dead Sea region, based on 1927 macroseismic data integrated with modern measurements.

Our new attenuation equation contains a term which quantifies solely lithological effects, whilst factors such as building quality, foundation depth, topography, earthquake directivity, type of fault, etc., remained out of the equation. Nonetheless, about 60% of the measured anomalous sites fit expectations and better fitting is achieved compared to other relevant 20   attenuation equations.

From a local point of view, this is the first time that an integration between historical data and modern seismic measurements improves the attenuation relation for the Dead Sea region. In the wider context, regions of low-to-moderate seismicity should use historical earthquake data together with modern measurements in order to better estimate the peak ground acceleration or the seismic intensities caused by future earthquakes. This integration will conceivably lead to a better understanding of 25   future earthquakes and improve maps of seismic risk.



# 1 Introduction

## 1.1 Site Response

Ground motion is controlled by a number of variables, including source characteristics, propagation directivity, near-surface geology, etc. Elastic properties of near-surface materials and their effects on seismic wave propagation are crucial for
earthquake and civil engineering, and in environmental and earth science studies.

Seismic waves are generated at the moment that an earthquake occurs. These waves are traveling and shaking the surface as they pass. The wave amplitude at the surface is affected by the mechanical properties of the rocks below. These rocks often consist of weathered rock over bedrock with a much higher seismic velocity. When seismic waves pass from a high-velocity layer to a low-velocity layer their amplitudes and duration can increase. Therefore, lithology is a crucial factor for estimating
site amplification, defined as the amplitude ratio between the surface layer and the underlying bedrock. Site amplification is due reverberation of the seismic waves in the upper layers according to acoustic impedance differences (Figure 1). The amplification, $A$, is proportional to the reciprocal square root of the product of the shear-wave velocity, $Vs$. (Eq. (1)) (Aki and Richards, 2002):

$$A \propto \frac{1}{\sqrt{V_s \rho}}, \tag{1}$$

where $\rho$ is the density of the investigated soil. Since density plays a minor role (Moro, 2015; Xia et al., 1999) the $V_s$ value can be used to represent site conditions.

The phenomena of site amplification as a result of soft sediments on top of hard bedrock is well known since the early days of seismology (Milne, 1898). Also, site-effects are well known and were investigated based on several major earthquakes: Mexico City 1985 (Singh et al., 1988), Armenia 1989 (Borcherdt et al., 1989), San Francisco 1989 (Hough et al., 1990), Los
Angeles 1994 (Hall et al., 1994) and Kobe (Japan) 1995 (Aguirre and Irikura, 1997).

The most widely used parameter in the classification of the soil response is the average shear-wave velocity in the uppermost 30 meters of sediment, the $Vs_{30}$. This parameter was accepted for site classification in the USA (National Earthquake Hazards Reduction Program - NEHRP), (Building Seismic Safety Council, 2001), also in Europe by the new provisions of Eurocode 8 (BSI, 2011), and in Israel it is accepted by the Standards Institute (The Standards Institution of Israel, 2013). The
value of 30 meters comes from the USA and European building codes where it was found empirically that this depth is directly proportional to deeper and shallower values (Boore et al., 2011). In Israel, there is not much data for this kind of correlation. Therefore, in this scenario, the Israel Standards Institute adopts the $Vs_{30}$ parameter.

In modern attenuation equations, also known as ground motion prediction equations (GMPE), coefficients are set from strong motion data, namely ground acceleration measurements. In the past, and in areas lacking the technology to record



earthquakes, it is impossible to directly measure the peak ground acceleration (PGA). Therefore, it is common to categorize historical earthquakes with seismic intensity scales that describe the damage at each site or area (Ambraseys, 2009; Guidoboni and Comastri, 2005)

**1.2 Jericho 1927 earthquake**

The 6.2M$_L$ July 11, 1927 Jericho earthquake (Ben-Menahem et al., 1976; Shapira, 1979) was the strongest and most destructive earthquake to hit the Holy Land during the past and current centuries. Furthermore, for the first time an earthquake with epicenter in the Holy Land was recorded by seismographs. The epicentral location was estimated at a few kilometers south of the Damia Bridge, which is 30 kilometers north of Jericho (International Seismological Summary – ISS Bulletin of 1927). A few decades later, new estimates have been calculated: Shapira et al (1993) calculate the epicenter to be

near Mitzpe Shalem. Zohar & Marco (2012) estimated the epicenter to be near the Almog settlement, about 30 kilometers north of Shapira's epicenter, and Kagan et al (2011), surmised that the source was somewhere on the Kalia fault which is located in the northern part of the Dead Sea graben, perpendicular to the main Dead Sea fault (Figure 2).

The damage from the earthquake was heavy, especially in places near the source, but not only there: In Nablus, located 70 km from the epicenter, 60 people were killed, 474 were injured, and more than 700 structures were destroyed, most of which

were built on soft sediments (Blankenhorn, 1927; Willis, 1928). By comparison, Jerusalem is only about 30 kilometers from the source and the damage there was much smaller, especially in property. However, in Mount Scopus and the Mount of Olives (neighborhoods in Jerusalem), the damage exceeded that in other parts of Jerusalem (Abel, 1927; Brawer, 1928). Other cities also suffered from this earthquake:  Tens of people were injured and even died, and hundreds of houses were ruined in Ramleh and Lod (Brawer, 1928). In addition, Jericho suffered significant damage, especially in terms of buildings

collapsing (Figure 3). The total number of victims was about 350-500 (Ambraseys and Melville, 1988; Amiran, 1951; Arieh, 1967; Ben-Menahem, 1991). Beyond the casualties, several environmental effects were reported: The Jordan river flow ceased near the Damia bridge for about 21.5 hours (Willis, 1928) and a one meter seiche wave was observed in the Dead Sea (Abel, 1927; Blankenhorn, 1927). Some evidence suggests that the earthquake was felt up to 700 kilometers from the epicenter (Ben-Menahem, 1991), although a different interpretation suggests this distance was only 300 kilometers

(Ambraseys and Melville, 1988).

Compiling historical evidence, Avni (1999) estimated the seismic intensities (MSK scale) at 133 different locations around Israel, Palestine, Jordan, Lebanon, Syria, and Egypt (Figure 4). Avni's basic attenuation equation yields an R$^2$ of about 0.26. Hough & Avni (2011) revised the attenuation equation for the Dead Sea region based on Bakun (2006):

$$MMI(M,d) = -0.64 + 1.7M - 0.00448d - 1.67\log(d)$$ , (2)

where *MMI* is Modified Mercalli Intensity (assumed to be equivalent to MSK), *M* is the magnitude and *d* is the distance from the epicenter.




Our main goal in this research is a tighter constraint on the attenuation equation for this event. This should allow us to examine whether this preliminary work coincides with our expectations of site amplifications and de-amplifications due to the lithology.

**2 Methods - Multi Analysis of Surface Waves (MASW)**

5 The MASW method is environmentally friendly, non-invasive, low-cost, rapid, robust, and provides reliable $Vs_{30}$ data (Miller et al., 2002). Multichannel records make it possible to separate different wavefields in the frequency and velocity domains. Fundamental and higher modes can be analyzed simultaneously, but generally, only the fundamental mode is used because among all the wave types it has the highest energy (Park et al., 1998).

The MASW method can be subdivided into three main steps: (A) Acquisition of experimental data, (B) signal processing to

10 obtain the experimental dispersion curve, and (C) inversion to estimate $Vs_{30}$ (Figure 5). The inverse problem consists of estimating a set of parameters that describe the soil deposit, based on an experimental dispersion curve. Inversion problems based on wave propagation theory cannot be solved in a direct way due to their non-linearity. Thus, iterative methods must be used where a theoretical dispersion curve is determined for a given layer model and compared to the previously obtained experimental dispersion curve (Ryden et al., 2004). $VS_{30}$ typically does not converge to one stable value. In other words, for

15 the same dispersion curve, one will get slightly different $Vs_{30}$ depending on the initial model.

We carried out the surveys with an array of 24 vertical geophones (R.T. Clark's geophones of natural frequency of 4.5 Hz) at equal intervals of 2-3 meters over a total length of 46-69 meters. In order to initiate the system, we used a five-kilogram sledgehammer pounding a twenty centimeter square aluminum plate at variable offsets of 5, 10, 15, 20, 25 and 30 meters

20 (both forward and reversed). The seismic data were recorded on a Geometrics Geode seismograph at a sampling rate of 8 kHz for 0.5-2 seconds (Table 1). For an acceptable Signal to Noise Ratio (SNR), we used the so-called "vertical stacking" approach, which is a summation of multiple synchronized repetitions of the test (usually five times).

Rayleigh wave dispersion curves were obtained by the MASW module of the RadExPro software in which the calculation procedure is based on a paper by Park et al. (1998). From all the dispersion curves that we picked for each site (Figure 6A),

25 we chose the smoothest and clearest dispersion image (Figure 6B) to compute the site's $Vs_{30}$ profile (Figure 6D). An inversion process then finds that shear-wave velocity profile whose theoretical dispersion curve is as close as possible to the experimental curve (Figure 6C). This procedure is done by Occam's inversion which is part of the MASW module of the RadExPro software. During this process, the Root Mean Square (RMS) error between the curves is minimized while maintaining the maximum model smoothness (Constable et al., 1987).





## 3 Results

Avni's (1999) original attenuation equation yields a $R^2$ of 0.26. Hough & Avni's (2011) revised equation, based on Bakun and Wentworth (1997), yields the same fit. Identifying the Kalia fault as the source location (Kagan et al., 2011) in Eq. (2) yields a fit of 0.35. The best fit of 0.38 is obtained using the Almog settlement as the epicenter (Zohar and Marco, 2012).

Accordingly, our analysis is based on this epicenter location. A scatter diagram of the distribution of all 133 sites for which Avni (1999) estimated seismic intensity, together with a prediction boundary of 60% from Eq.(2), highlight the sites that were amplified and de-amplified (Figure 7).

### 3.1 MASW surveys

From these 24 surveys, we succeeded in extracting $Vs_{30}$ for 19 of 20 sites (the Hartuv data were too noisy for interpretation)
(Table 2).

## 4 Discussion

### 4.1 Survey locations and validation

The decision as to where exactly each survey would take place was based on Avni's thesis (Avni, 1999). If the location was not sufficiently clear we checked the reference given by Avni. In most cases, there was evidence of specific damaged
buildings. We tried to locate those buildings while looking at historical maps (1927-1945). Unfortunately, most sites were located inside urban areas where we could not execute the seismic surveys. Therefore, we surveyed in open areas as near as possible to the referenced damage zones. To validate our results, we compared them with the nearest (up to three kilometers distance) data from The Geophysical Institute of Israel (GII) and a reasonable fit was achieved (Figure 8). However, this comparison is a bit tricky because $Vs_{30}$ results for two sites distant three kilometers or much less could be significantly
different, as shown in Figure 9. Remembering that $Vs_{30}$ enters a logarithmic term, we find our approach potentially useful.

### 4.2 Velocity model

All models were considered as a stack of homogeneous linear elastic layers, neglecting lateral variations in soil properties. The number of unknowns for a layered model, when considering only shear-wave velocity, is three for each layer: density, thickness, and one elastic constant. Therefore, the number of unknowns is 3n-1 (where n represents the number of layers).
The change in density with depth is usually small in comparison to the change in shear modulus and is normally neglected (Park et al., 1997). Therefore we set the density to 2000 [kg/m$^3$] for all layers in all our sites and thus the number of unknowns decreases to 2n-1.





### 4.2.1 Number of layers & layer thicknesses

The resolution of surface wave surveys decreases with depth. Thin layers are well resolved when they are close to the surface, whereas at great depth, the resolution is limited and only large changes can be detected. The reduction of the sensitivity with depth results in a loss of resolution, or in the ability to identify the properties of thin layers. Thus, these features cannot be accurately resolved (Foti et al., 2014). Regardless of the number of the layers of the site, $Vs_{30}$ is almost the same in each case (Figure 10). For those reasons, as well as the lack of density information, we did not restrict each model to a specific number of layers. Without boreholes and lithostratigraphic data, which is the case in our work, a good rule of thumb is to assume layer thicknesses increasing with depth, to compensate for the decreased resolution with depth, which is an intrinsic shortcoming of surface wave testing (Foti et al., 2014).

### 4.2.2 Depth of investigation

We used a five-kilogram sledgehammer and summed up five strikes. In some sites this type of source is insufficient to determinate a shear-wave profile down to 30 meters. In addition, at some sites we were not able to spread the geophones at intervals of more than two meters which limited the length of the seismic line. This length probably excludes longer wavelengths which limits the depth of investigation. Lastly, as the shear-wave velocity of the lowest frequency is higher - more data is available for deeper layers. Therefore, the penetration depth will decrease in areas with low shear-wave velocity. For instance, if we can clearly detect a phase velocity of about 200-300 m/sec at 5 Hz, we can roughly estimate a depth of investigation of approximately 12-20 meters according to the following equation:

$$Z = \frac{\left(\dfrac{Velocity_{f_{min}}}{f_{min}}\right)}{n} \quad , \tag{3}$$

where $n$ equals 2-3 (Foti et al., 2014; Moro, 2015). In other words, this equation emphasizes that the depth of investigation is about a half to a third of the largest wavelength observed.

### 4.3 A new attenuation equation

In the present case of the 1927 earthquake, the sources of the data are mostly historical documents and not measurements. This makes it difficult to quantify site response into a single equation. In the practical modern attenuation relation, $Vs_{30}$ is a crucial index. A term that depends on $Vs_{30}$ has been constrained for several large data sets (Abrahamson et al., 2014; Boore et al., 1997; Campbell and Bozorgnia, 2008). We chose the Boore et al. (1997) attenuation equation (Eq. (4)) in order to emphasize site response.

$$\ln Y = b_1 + b_2(M-6) + b_3(M-6)^2 + b_5 \ln(r) + b_v \ln\left(\frac{Vs}{V_A}\right), \tag{4}$$



where $Y$ is the ground-motion variable (peak horizontal acceleration or pseudo-acceleration response in g), $M$ is the moment magnitude, $r$ is the epicentral distance in kilometers, $V_A$ and all $b$ are frequency dependent coefficients to be determined. By adding Boore et al.'s Vs term to Hoguh and Avni's attenuation equation (Eq. (2)), we suggest a new equation for the Dead Sea region:

$$MMI = -0.64 + 1.7M - 0.00448d - 1.67\log(d) + C_4 \ln\left(\frac{Vs_{30}}{Vs_{refernce}}\right),$$
(5)

where $Vs_{refernce}$ is the shear-wave velocity of the bedrock and $C_4$ is an adjustable constant. Optimizing this constant to our data yields the final equation:

$$MMI = -0.64 + 1.7M - 0.00448d - 1.67\log(d) - 1.8\ln\left(\frac{Vs_{30}}{760}\right),$$
(6)

We adopt here the value of $V_A$ from Boore's equation (equation 2) as representing $Vs_{refernce}$. According to most national

standards, including the one in Israel (SI #413), the reference bedrock shear-wave velocity is set equal to 760 m/sec for all frequencies in the entire region. With these coefficients, 58% or 11 of 19 sites, were amplified or de-amplified as we expected. Based on the new attenuation equation (Eq. (6)) we reduced the site-effects (Figure 11) and compare the fitness of our attenuation curve with those of Avni (1999) and Hough & Avni (2011) (Figure 12). The new attenuation equation fits the data somewhat better for all four epicenters.

Boore et al.'s equation (Eq. (2)) is restricted for use only for earthquakes of magnitude 5.5-7.5 and epicentral distance up to 80 kilometers. After lithological corrections, sites located up to 70 kilometers from the epicenter are well predicted: the entire population of six anomalous sites shifted to the prediction boundary. On the other hand, sites located farther than 70 kilometers from the epicenter converge into the prediction boundary to a lesser extent (five sites of thirteen which are 38%) (Figure 11). This observation is consistent with Boore's restriction.

For the entire distance range (up to 250 km) the $Vs_{30}$ corrections leave 42% sites out of the prediction boundary (eight of nineteen sites). Seismic intensities in all these eight sites are underpredicted by the attenuation equation (Eq. (2)) (Figure 11). We expect that $Vs_{30}$ at these sites will be higher than 760 m/sec in order to obtain de-amplification. However, our results show the opposite effect - these eight sites are characterized by lower $Vs_{30}$ which drive amplification. This can be caused by the fact that measurements were taken over agricultural fields, of which the upper layers (the first few meters) are

characterized by low shear-wave velocity, decreasing the average Vs. Another reasonable explanation is that we did not succeed in extracting the average shear-wave velocity down to 30 meters and perhaps we missed some high-velocity shear-wave layers at deeper layers. In such cases, we constrain the last layer to be thicker in order to estimate $Vs_{30}$ for all our surveys.



## 5 Conclusions

In this research, we investigated site amplification and de-amplification around Israel. According to previous studies (Aki, 1988; Boore, 2003; Borcherdt, 1994; Field and Jacob, 1995; Joyner and Boore, 1988) the local lithology can amplify or de-amplify wave amplitude. The commonly used modern seismic method – MASW – allows the extraction of Vs profiles at 20

sites reportedly damaged by the 1927 $M_L6.2$ earthquake. We use these profiles to update the attenuation equation for the Dead Sea region by including the $Vs_{30}$ term.

According to this new equation, 11 sites, which constitute 58% of our measured samples, move into the 60% prediction boundary. This suggests that the prediction boundary actually encompasses over 80% of the macroseismic observations. This fit is better than any available attenuation equation for the Dead Sea region (Figure 11). However, as we have used only 19

sites, we should consider further research and provide wider results. Although our final equation (Eq. (6)) shows amplification and de-amplification depending on $Vs_{30}$, it does not take into consideration any other factor such as building quality, foundation depth, topography, earthquake directivity, type of fault etc. Obviously, for better results we must use more methods and jointly invert some other seismic data such as: refraction (S and P waves), Horizontal to Vertical Spectral Ratio (HVSR), MASW of the transverse component of Love waves, MASW of the radial component of Rayleigh wave,

Extended Spatial Auto-Correlation (ESAC), etc.

Despite the scarcity of data, this is the first time that an integration of historical data with modern seismic measurements improves the attenuation relation. In order to better estimate the peak ground acceleration or the seismic intensities that will cause by future earthquakes, attenuation relations are necessary worldwide, especially in areas characterized by high seismicity. Some of the regions of low to moderate seismicity have rich historical earthquake data. The integration of

historical data with modern measurements will lead to a better understanding of future earthquakes.

*Acknowledgments.* We thank the Neev Center's facility and its students. We are especially grateful to Dr. John K. Hall who founded the Center, for his ongoing support. We are grateful to the Helmholtz Association of German Research Centers for funding this research. We thank Prof. Moshe Reshef for comments and suggestions on an earlier draft and Prof. Ran Bachrach for valuable advice. We acknowledge

the contribution of Prof. Michael Weber and the geophysical deep sounding section at GFZ. Finally, we thank Amit Ronen for his assistance.



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

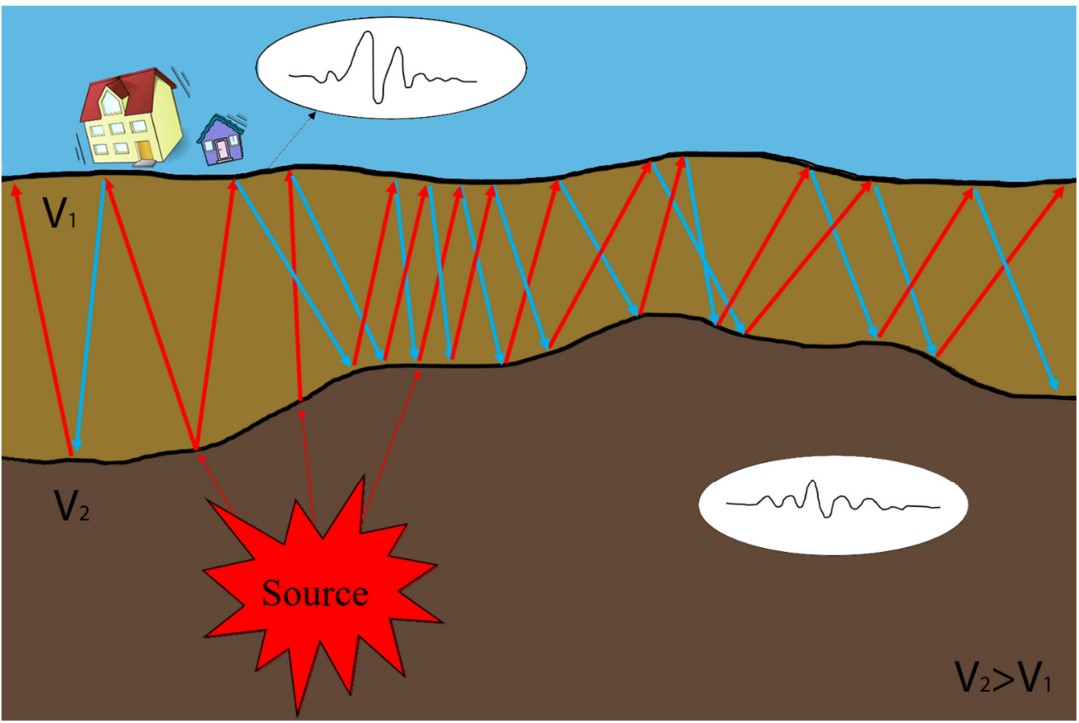

**Figure 1: Schematic view of site amplification due to reverberations. Seismogram at the surface shows amplification in comparison**
10   **to the seismogram located over the bedrock.**



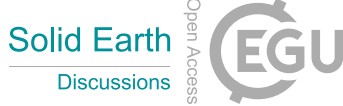

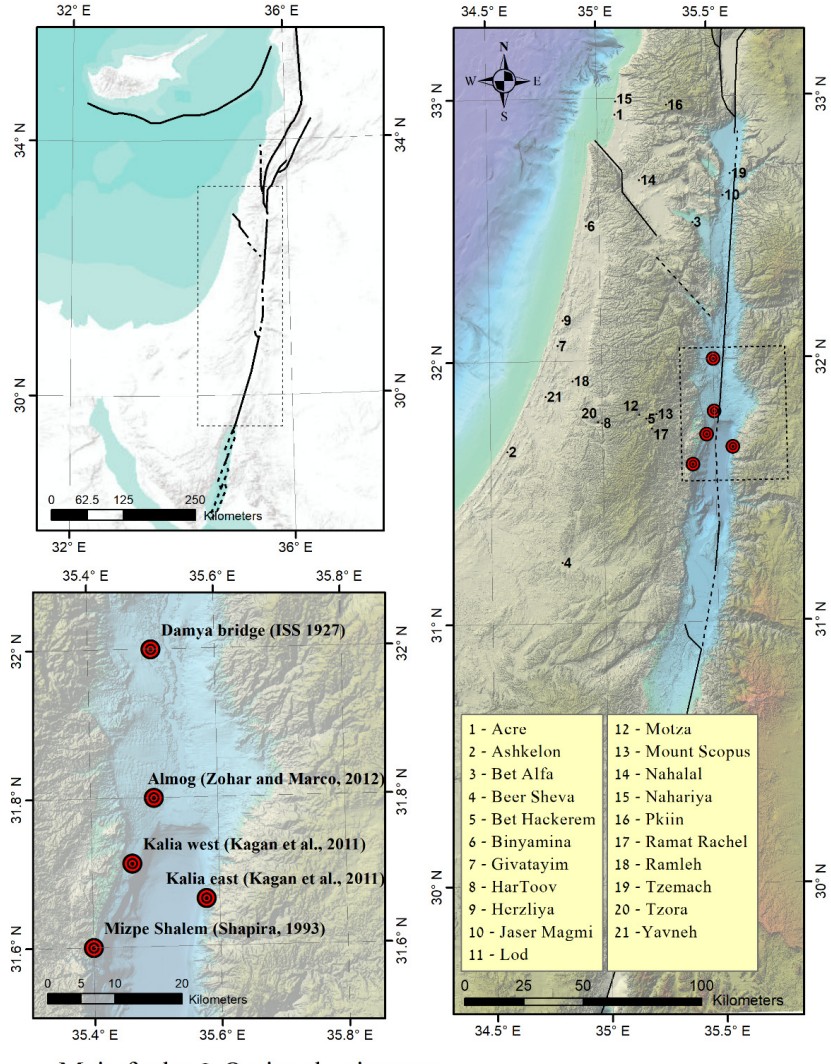

**Figure 2: A) Middle East area. B) Optional epicenters for the 1927 earthquake event with all sites that were investigated placed over the DTM map made by John K. Hall. C) Location of the optional epicenters.**




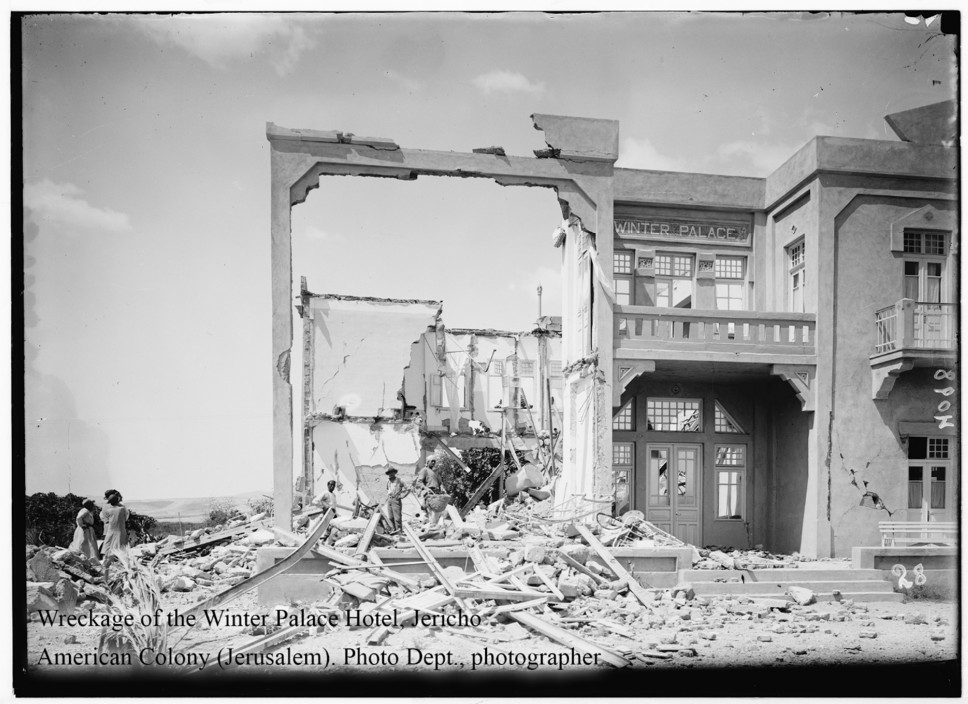

**Figure 3: Wreckage of the Winter Palace Hotel, Jericho, after the 1927 earthquake. American Colony (Jerusalem). Photo Dept., photographer.**



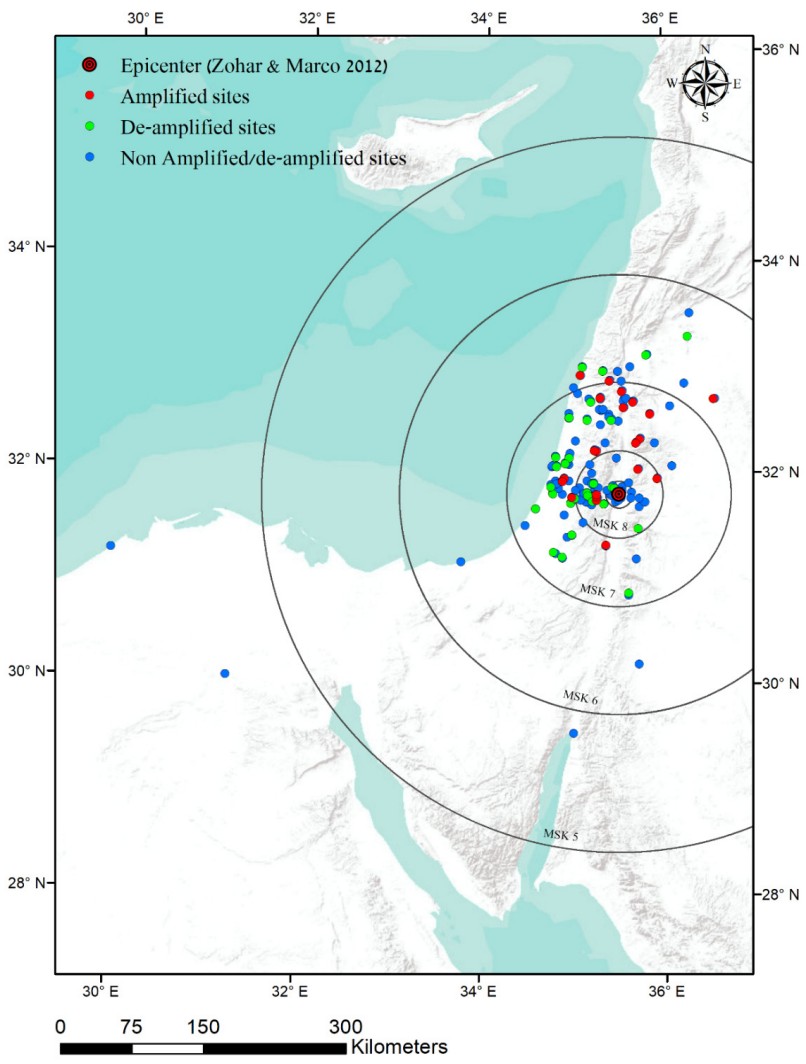

Figure 4: Isoseismal map. The epicentral locations in red and black circles. Red and yellow dots are suspect amplified or de-amplified sites (respectively). Blue dots are sites which have MSK values expected from the attenuation equation (with 60% prediction boundary).





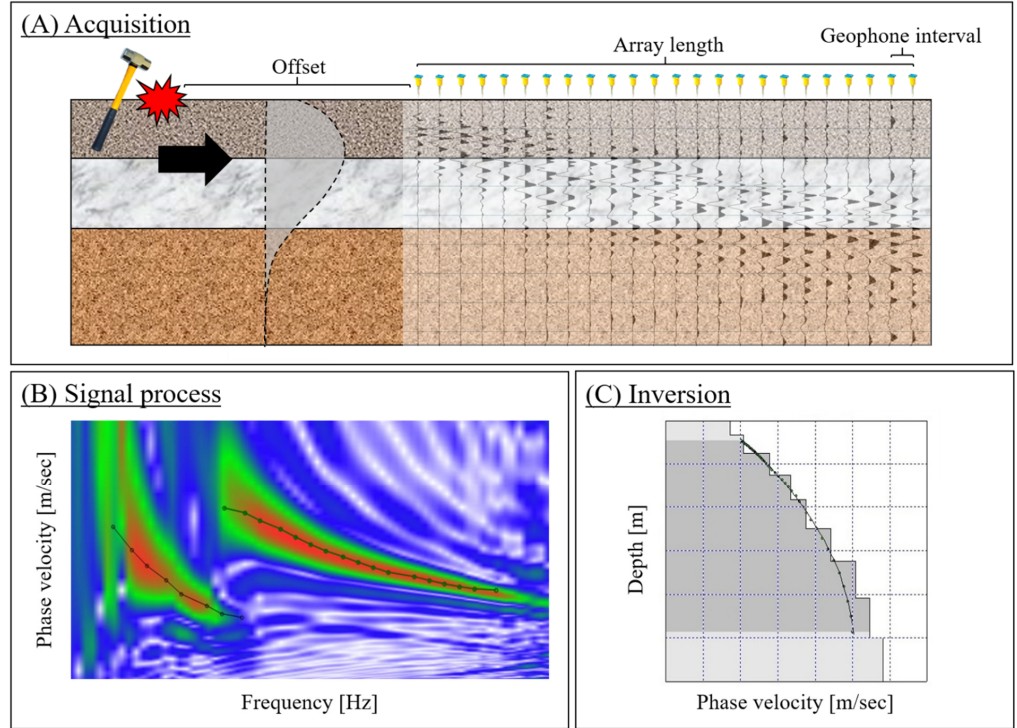

**Figure 5: Multichannel Analysis of Surface Waves (MASW) technique: A. Acquisition – Using a sledgehammer as an artificial source and a linear array of geophones that receives all wavelets. B. Signal process – A fundamental mode and first higher mode over the dispersion image. C. Inversion – Final $V_s$ profile which fits the best to the dispersion curve.**



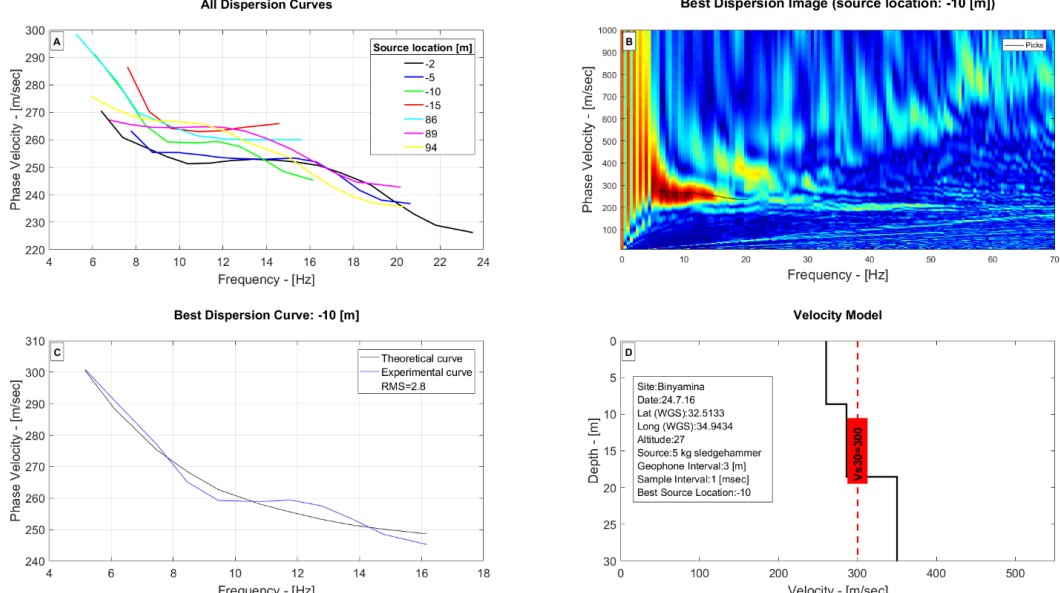

**Figure 6: Data processing (example from Binyamina): A – Seven dispersion curves of each offset. B – The smoothest and clearest dispersion image (Offset -10). C – Best dispersion curve and a suitable theoretical one. D – Shear-wave velocity model and $Vs_{30}$ value.**





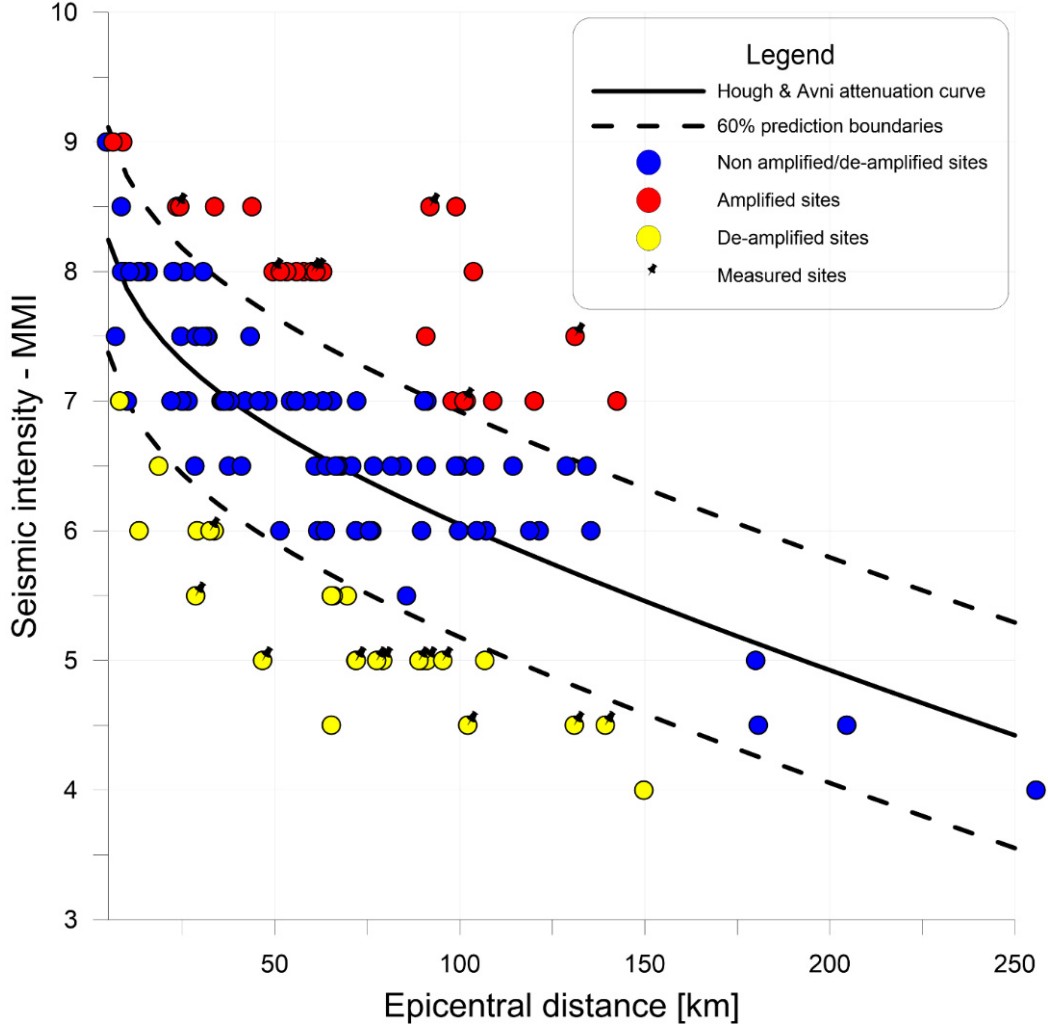

**Figure 7: Avni's seismic intensity (MMI) estimates of all 133 sites. Distance is corrected according to Zohar & Marco epicenter. Red and yellow dots suspected amplified or de-amplified sites (respectively). Sites with pin are sites where we measured the Vs profile. Blue dots are sites which have MSK values expected from the attenuation equation (with 60% prediction boundary).**





**Acquisition parameters**

| | |
|---|---|
| **Number of geophones** | 24 |
| **Geophone spacing** | 2-3 meters |
| **Array length** | 46-69 meters |
| **Sampling rate** | 8 kHz |
| **Record length** | 0.5-2 second |
| **Receivers** | 4.5 Hz vertical |
| **Source** | 5 kg hammer |

Table 1: Acquisition parameters.

| ID | Site | $Vs_{30}$ [m/sec] | Mean $Vs_{30}$ [m/sec] |
|---|---|---|---|
| 1 | Acre | 240 | 240 |
| 2 | Ashkelon | 490 | 490 |
| 3 | Be'er Sheva | 360 | 360 |
| 4 | Beit Hakerem | 1450 | 1450 |
| 5 | Beit Alfa | 230 | 230 |
| 6 | Binyamina | 300 | 300 |
| 7 | Givatayim | 410 | 410 |
| 8 | Herzliya | 320 | 320 |
| 9 | Jasar-Majami | 260 | 260 |
| 10 | Lod 1 | 320 | 340±20 |
| 11 | Lod 2 | 360 | |
| 12 | Motza 1 | 1030 | 940±90 |
| 13 | Motza 2 | 850 | |
| 14 | Mt. Scopus | 580 | 580 |
| 15 | Nahalal | 380 | 380 |
| 16 | Nahariya | 830 | 830 |
| 17 | Peqi'in | 1330 | 1330 |
| 18 | Ramleh 1 | 320 | 350±30 |
| 19 | Ramleh 2 | 380 | |
| 20 | Tzemach 1 | 220 | 245±25 |
| 21 | Tzemach 2 | 270 | |
| 22 | Tzora | 310 | 310 |
| 23 | Yavneh 1 | 540 | 450±90 |
| 24 | Yavneh 2 | 360 | |

**Table 2: MASW results.**

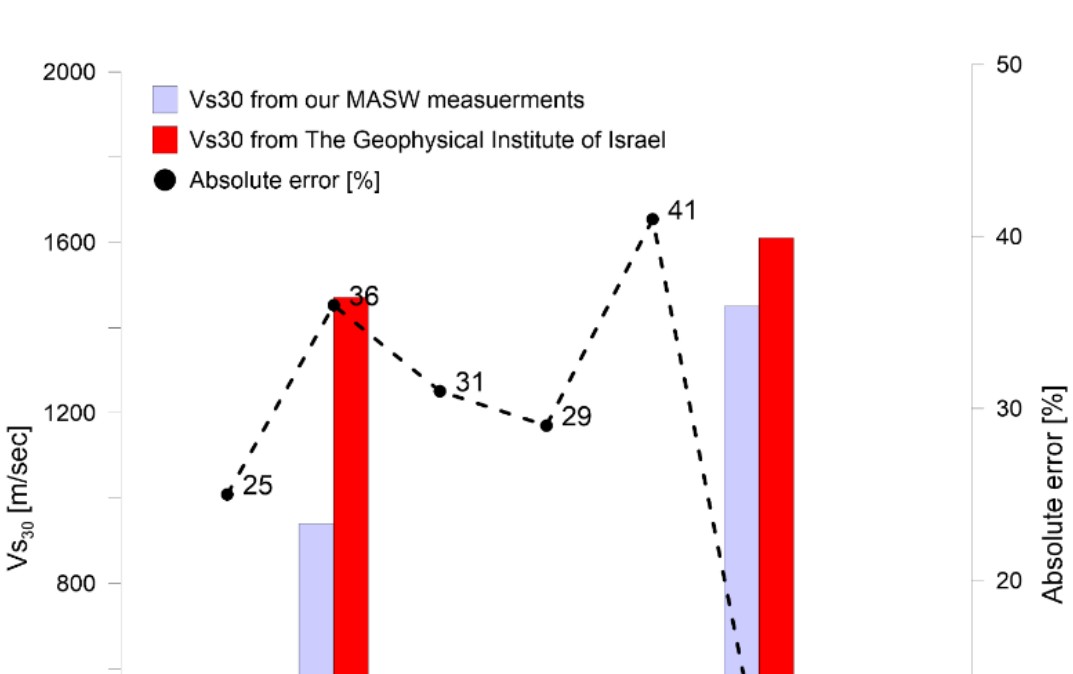

**Figure 8: Comparison between our Vs₃₀ results and those calculated from GII's reports.**



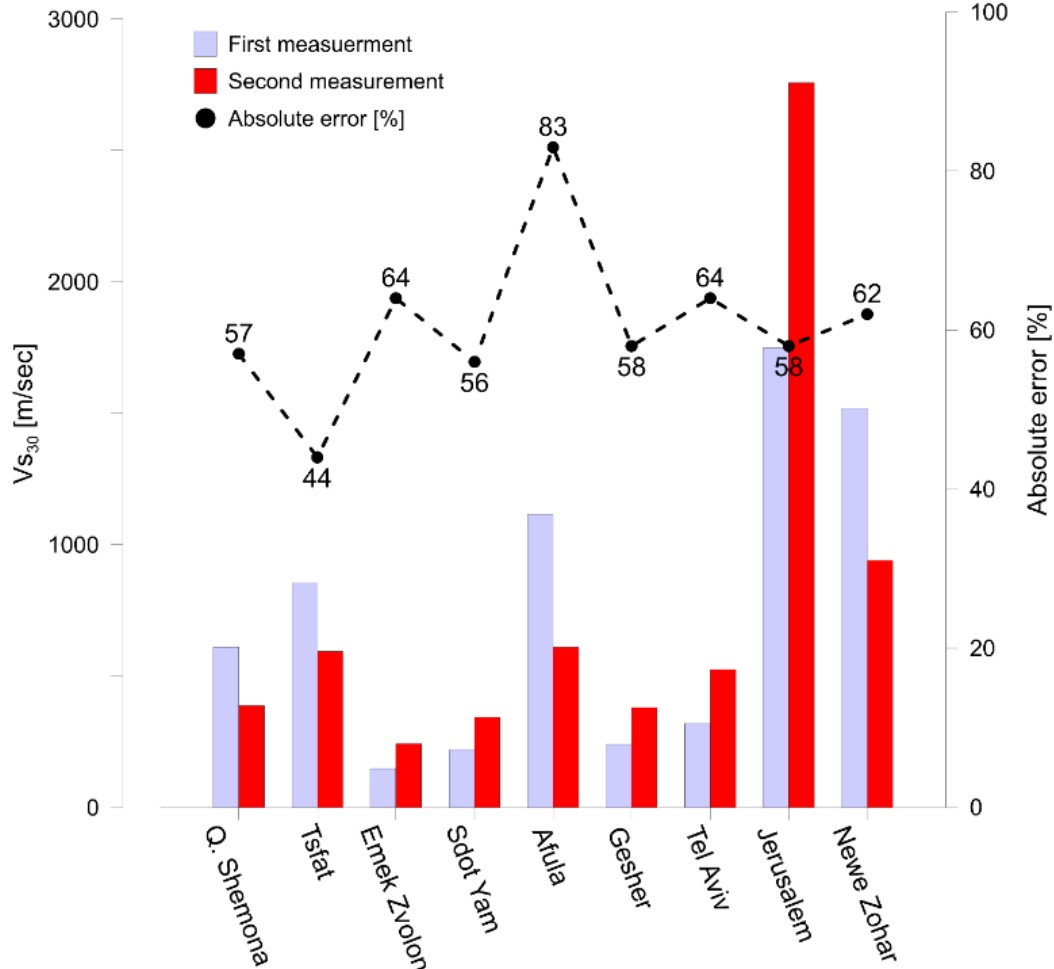

**Figure 9: Comparison between GII's closest measurements (up to 550 meters).**





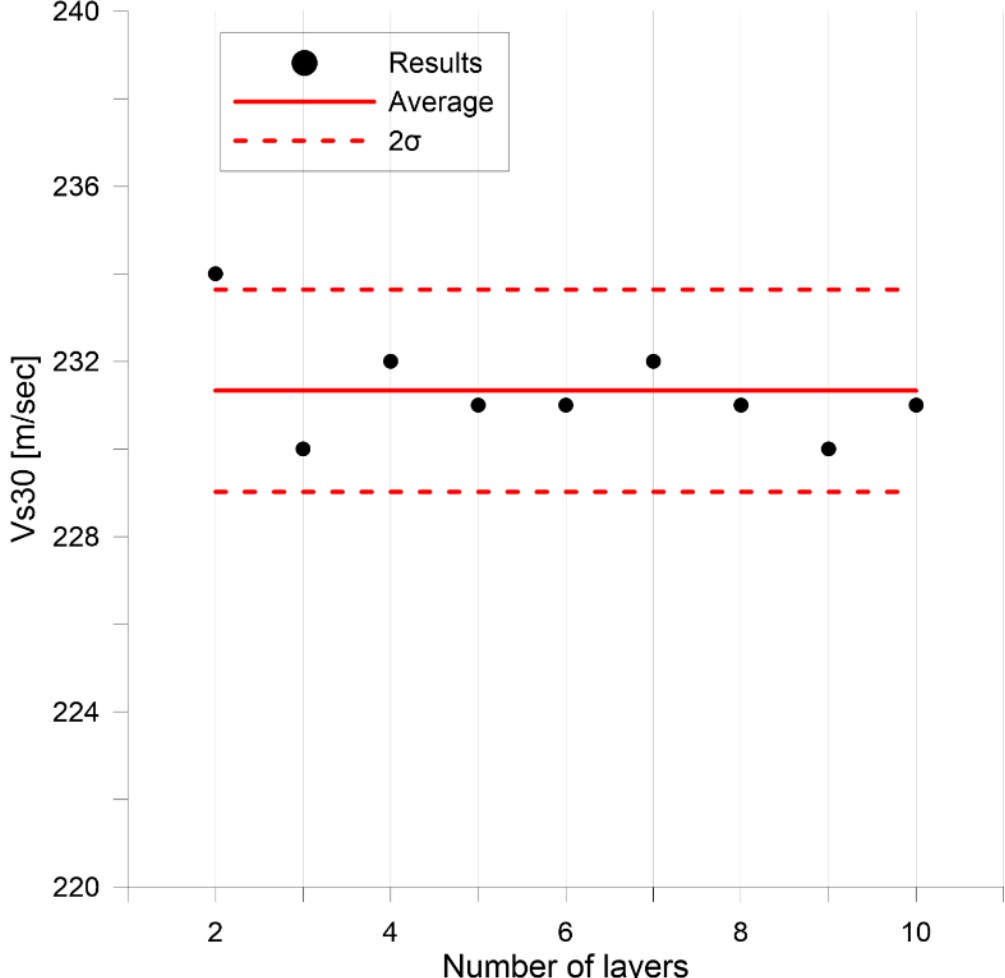

**Figure 10: Vs₃₀ as a function of a number of layers.**



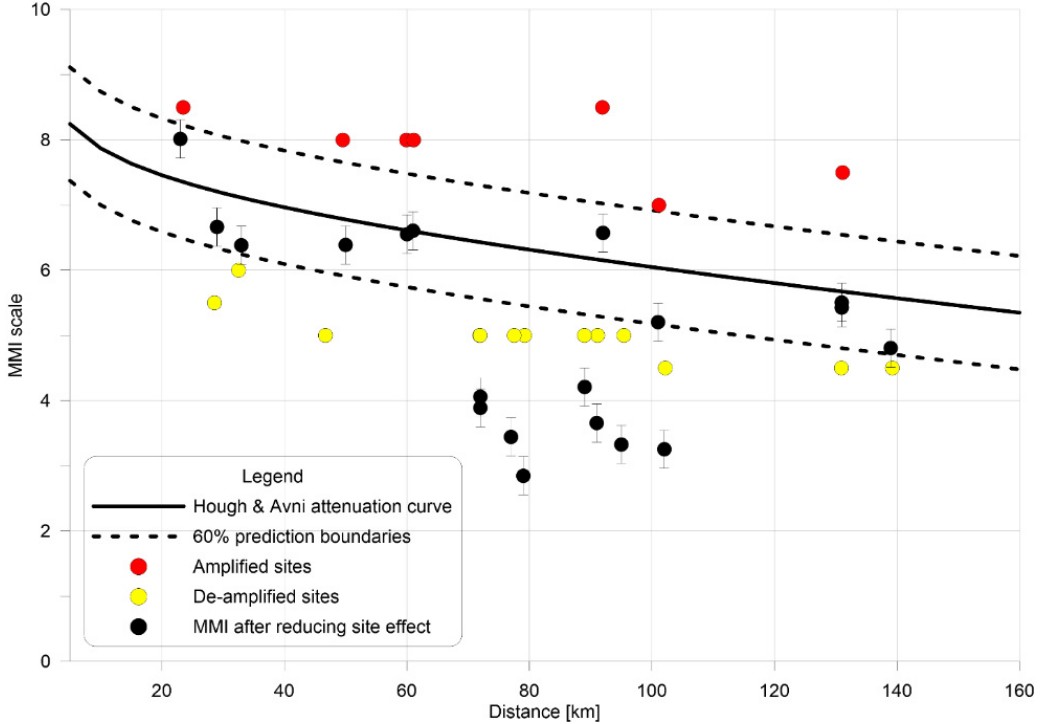

**Figure 11: Red and yellow dots suspect amplified or de-amplified sites (respectively). Black dots with error bars due to 15% Vs uncertainty represent the MMI after reducing site effect.**





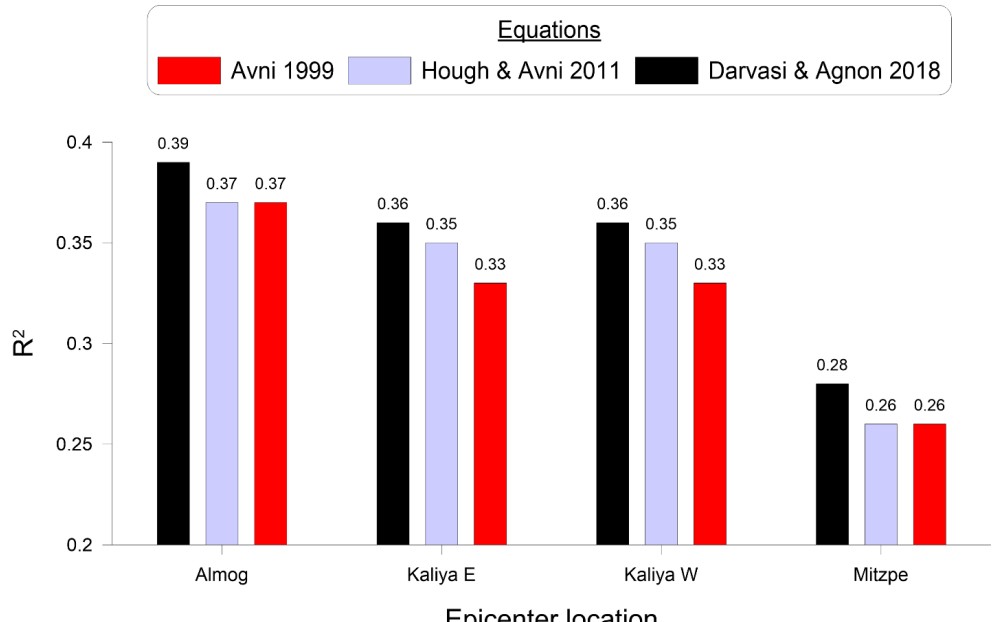

Figure 12: R$^2$ comparison of all epicentral locations and attenuation equations.

