# Peer review of "Calibrating a New Attenuation Curve for the Dead Sea Region Using Surface Wave Dispersion Surveys in Sites Damaged by the 1927 Jericho Earthquake"

_Solid Earth, 2018_

## Referee Comment (RC1) · Anonymous Referee #1 · 2 Oct 2018

Review of manuscript se-2018-52 titled Calibrating a new attenuation curve for the Dead Sea region using surface wave dispersion surveys in sites damaged by the 1927 Jericho earthquake by Darvasi Y. and Agnon, A.

The above-mentioned manuscript describes a study where an attenuation relation for the Dead Sea was calibrated, using intensity data from the 1927 north Dead Sea earthquake, along with shear wave velocity measurements from the sites where intensity was documented. An former intensity GMPE for the Dead Sea was taken, and a site term was added to it, to account for site response effects. The new GMPE shows a

better fit for 60% of the data points.

General comment

The topic this manuscript addresses is important, especially for regions where strong-motion data is scarce, like Israel. Finding a way to utilize historical data can contribute much to seismic hazard estimations in those regions. However, one major question that arises from this manuscript is what is the contribution of the new GMPE? The original equation by Hough and Avni was based on a single event and about 130 data points. This manuscript attempts to improve their GMPE and include a site term using only 19 data points, while obtaining a better fit for only 60% of them. In the opinion of the authors – can this GMPE be used for intensity prediction in the future? I feel the step taken in this manuscript is too small – the dataset it is based on is small, no sensitivity analyses were performed and no validations of the GMPE were performed. Can the authors compare macro-seismic data from other historical and perhaps instrumental earthquakes to their GMPE and discuss the results?

Other more specific comments:

The authors took a formerly derived GMPE, with its regression and constants, and simply added another term, regressing only for its constant. Such an action means the authors think that the magnitude, attenuation, geometrical spreading and site response are all independent. That is not fundamentally wrong, but should be stated. If the authors would perform the regression on all variables, they probably would get slightly different values.

The authors adopt 760m/s as the Vsreference for their new GMPE, although this value is probably not suitable for Israel, as the Judean group is close to the surface in many regions, and its shear wave velocity is higher. Perhaps a sensitivity analysis for this value would come up with a value better fitting the data, and would contribute to the point the authors are trying to make.

After developing their own site term, the authors try to explain why 8 points did not "converge into the prediction boundary" of their equation. They explain that Boore et al.'s GMPE is constrained to a distance of 70 km, and some their data is further away. This argument is weak, as they did not adopt Boore et al.'s equation, but Hough and Avni's intensity equation, and only Boore's site term, therefore using their distance constraint does not seem to explain the misfit. Also – 5 sites beyond 70 km did converge to their GMPE.

The authors fail to discuss or at least mention other research regrading this exact earthquake, such as Kadmiel et al. (2015, SCEC).

I think it would be interesting if the authors try to use the data from Zohar and Marco (2011), which attempted to correct the intensities of the 1927 EQ to site conditions, such as surface geology, slope and construction. Zohar and Marco's work should at least be mentioned, as they did try to deal with the site response biasing the intensity reports.

Technical comments

The manuscript should be English and grammar proofed before re-submitted.

The magnitude of the earthquake in the abstract and in the Introduction is different. Although the difference is small- please pick one.

In the abstract and several more instances MASW is explained as Multi Analysis of Surface Waves, please correct to Multichannel Analysis of. . .

In the abstract and in several more instances the authors state that (line 15-16) ". . .based on 1927 macroseismic data integrated with modern measurements". This phrasing is somewhat misleading, since it hints that new seismic data is being used, where I believe the authors are referring to the velocity profile measurements they performed.

Abstract – line 25 –seismic hazard and not risk.

[Figure]

Interactive
comment

Introduction – some opening paragraph is missing.

Page 2 line 3 – how about source distance?

Page 2 line 10 – that is one definition of what amplification is, I think the authors should mention the other, common definition, relative to a near rock outcrop.

Page 2 lines 10-11 – site amplification can be attributed to many factors, such as basin effect, focusing effects, topography and on and on, and not only due to "reverberation of the seismic waves in the upper layers according to acoustic impedance differences". The decrease in shear wave velocity when getting closer to the surface is alone a reason for amplification of magnitudes, without any resonance in the upper layers.

Figure 1 is somewhat "overcrowded".

Page 2 lines 17-20 – this should come earlier in this section.

Page 2 lines 26-27 – several claims were made as to how is Vs30 suitable for Israel (i.e. Zaslavsky et al. 2012, Natural Science).

Where is the Hough and Avni (2011) reference in the list?

Vs30 notation is not consistent throughout the manuscript.

Is the equation of Hough and Avni based on Bakun (2006) (page 3 line 28) or Bakun and Wentworth (1999) (page 5 line 2-3) or both, and where are these references in the list?

Page 50 line 6 – what do the authors mean by 60% prediction boundary? Is that the boundary for which 60% of the data is included within? Unclear.

I'm not sure that section 4.1-4.2 belong in the discussion section. They seem to be more suitable in the methods section, since they are part of the modeling method. I don't think that anything novel is presented in these subsections.

Page 5 lines 17-18 – what kind of data from the GII was yours compared to? This is

very unclear. Is it borehole data? Refraction? HVSR? Is there a reference to the data?

Figure 10 – for which site are these results?

Page 7 line 3 – Hough's name is spelled wrong. Also the citing is not complete.

Page 7 line 9 – Boore's equation is equation (4) and not (2). Also citing not complete.

Figure 11 – the legend is not clear – does the curve represent the new GMPE, with the site term? Also I believe there is no need to use different symbols for amplified and de-amplified, they can be put in the same category as "MMI before site correction" or similar. Same goes for figure 7 – there is no need to use different colors for the symbols, and the use of the term "amplified" or "de-amplified" is not accurate – amplification is relative to reference rock conditions, and here I believe you mean the sites were "amplified" comparing to the GMPE.

page 7 line 15 – again Boore's equation is #4.

It would seem to be useful if the authors use index numbers to identify their measurement locations in the figures, as they are numbered in table 1, and also in table 1 include the epicentral distance of the measurements. This will allow the reader to understand the statements they make regarding the misfit of the different locations.

---

## Referee Comment (RC2) · Anonymous Referee #2 · 18 Nov 2018

Review of manuscript from Darvasi and Agnon: Calibrating a New Attenuation Curve for the Dead Sea Region Using Surface Wave Dispersion Surveys in Sites Damaged by the 1927 Jericho Earthquake

This manuscripts contributes to the applied research tasks of attenuation curve setup for regions suffering earthquake hazard. These are needed to give better estimates on ground motion in case of an event. In the lack of strong motion data in the region under consideration, both historical and MASW experiment datasets are evaluated in the Dead Sea region.

[Figure]

A major concern I see for this manuscript is the determination of dispersion curves and values from the experiments that are needed to establish the adapted attenuation function. Data are not shown in Chapter 3, and also the derivation of results and uncertainties are not given. This needs to be provided to the reader. As given here, one is not convinced. In the same line, the optimization of the constant (p.7, line 6 ) is only mentioned but not explained. These points must be better elaborated and proven in the methodical section. Furthermore, a discussion of the directivity of amplification in entirely missing.

General language: in parts very colloquial and not scientific/precise enough. Phrasing like somewhat, incomplete referencing, mentioning of authors without year of publication etc. might give an impression of rushed writing, that must be thoroughly revised.

Abstract: be more precise, what are specific results ?, this is rather an intro plus technical description.

Introduction: description of formula 1 is incomplete in text.

Methods: it is not clear if the term offset is used properly. It rather seems to mean profile distance in meters ?

Results: this chaper is completely inappropriate, because no data is introduced, nothing described, and no workflow given to the reader. Thereby, it is not possible to judge quality of results and be confidential in the outcome.

Discussion: some parts like velocity determination would belong to the results. The remaining discussion should also include a discussion of comparing achieved results here with other authors methods and workflows, not for the region alone but especially seen in worldwide literature.

Most of the figures are only mentioned in the text but need proper description there, too.

Figure 1: could be less scetchy. Figure 2: ...map made by JKH -> give proper reference or include him as author, if considerable work was done. Figure 8+9: correct legend spelling. Figure 9: what is second measurement ?, this is unexplained and not understandable. Figure 11: incomplete references in legend, caption:....dots mark suspected.....

In summary, this paper can grow to being worth published, but it is at the border and needs thorough major revision.

---

## Author Response (AR1)

**Reviewer #1**

We thank the reviewer for pointing out many importing issues. In order to be clear, the response structure is in the prescribed sequence:

- (1) Comments from Referee
- (2) Author's response
- (3) Author's changes in the manuscript
- (1) One major question that arises from this manuscript is what is the contribution of the new GMPE? Can this GMPE be used for intensity prediction in the future?
- (2) We thank the reviewer for pointing this out. The new GMPE is essential to emphasize how shear wave measurements are important in Israel and also in every modern GMPE. This work is preliminary. Therefore, the main contribution will be after collecting more data regarding additional earthquakes. For now, according to the new equation, 11 sites, which constitute 58% of our measured samples, move into the 60% prediction boundary. This suggests that the prediction boundary actually encompasses over 80% of the macroseismic observations. Hence, on the one hand, statistically, this new GMPE can be used. But on the other hand, in order to be a part of any standard, it should base on more data and also includes additional terms and not only a site condition term.
- (3) We added a sentence in the conclusion section to clarify these points
- (1) No sensitivity analyses were performed.
- (2) We thank the reviewer for the point and we include the sensitivity tests in the new version.
- (3) See page 8 (section 4.2). Also, fig11
- (1) No validations of the GMPE were performed.
- (2) At the moment a moderate earthquake will occur in the area we will be able to validate the GMPE.
- (1) Can the authors compare macro-seismic data from other historical and perhaps instrumental earthquakes to their GMPE and discuss the results?
- (2) This work focuses on the 1927 event and it is part of wider research which extends to additional potentially earthquakes. Certainly, comparing macro-seismic data from other historical events will enhance the attenuation relation and that is the future plan. Please keep in mind though that 1927 was the only destructive earthquake for which we have relatively reliable data.
- (1) The authors took a formerly derived GMPE, with its regression and constants, and simply added another term, regressing only for its constant. Such an action means the authors think that the magnitude, attenuation, geometrical spreading and site response are all independent. That is not fundamentally wrong, but should be stated.

- (3) Had been stated in the new manuscript.
- (1) The authors adopt 760m/s as the Vs reference for their new GMPE, although this value is probably not suitable for Israel, as the Judean group is close to the surface in many regions, and its shear wave velocity is higher.
- (2) We thank the reviewer for this point.
- (3) This part had been corrected.
- (1) After developing their own site term, the authors try to explain why 8 points did not "converge into the prediction boundary" of their equation. They explain that Boore et al.'s GMPE is constrained to a distance of 70 km, and some their data is further away. This argument is weak, as they did not adopt Boore et al.'s equation, but Hough and Avni's intensity equation, and only Boore's site term, therefore using their distance constraint does not seem to explain the misfit. Also – 5 sites beyond 70 km did converge to their GMPE.
- (2) You are definitely right about using Boore et al.'s equation.
- (3) We dropped out this explanation from the new manuscript.
- (1) The authors fail to discuss or at least mention other research regarding this exact earthquake, such as Kadmiel et al. (2015, SCEC). I think it would be interesting if the authors try to use the data from Zohar and Marco (2011), which attempted to correct the intensities of the 1927 EQ to site conditions, such as surface geology, slope and construction. Zohar and Marco's work should at least be mentioned, as they did try to deal with the site response biasing the intensity reports.
- (2) We mentioned the significant work of Zohar and Marco (2012 instead of 2011) and used their modern epicenter location for our calculation.
- (3) We added a comment about Kadmiel work.
- (1) The manuscript should be English and grammar proofed before re-submitted.
- (2) The original manuscript was English and grammar proofed
- (3) The new manuscript is English and grammar proofed
- (1) The magnitude of the earthquake in the abstract and in the Introduction is different. Although the difference is small- please pick one.
- (3) Had been corrected in the new manuscript.
- (1) In the abstract and several more instances MASW is explained as Multi Analysis of Surface Waves.
- (2) Some of the authors use it.
- (3) Had been corrected in the new manuscript.

- (1) In the abstract and in several more instances the authors state that (line 15-16) "based on 1927 macroseismic data integrated with modern measurements". This phrasing is somewhat misleading, since it hints that new seismic data is being used, where I believe the authors are referring to the velocity profile measurements they performed.
- (2) Comment accepted.
- (3) We redefined it in the new manuscript.
- (1) Abstract line 25 –seismic hazard and not risk.
- (2) Comment accepted.
- (3) Had been corrected in the new manuscript.
- (1) Introduction some opening paragraph is missing.
- (2) Comment accepted.
- (3) We added an intro section to the manuscript
- (1) Page 2 line 3 how about source distance?
- (2) Comment accepted.
- (3) Had been corrected in the new manuscript.
- (1) Page 2 line 10 that is one definition of what amplification is, I think the authors should mention the other, common definition, relative to a near rock outcrop.
- (2) Comment accepted.
- (3) Had been changed in the new manuscript.
- (1) Page 2 lines 10-11 site amplification can be attributed to many factors, such as basin effect, focusing effects, topography and on and on, and not only due to "reverberation of the seismic waves in the upper layers according to acoustic impedance differences". The decrease in shear wave velocity when getting closer to the surface is alone a reason for amplification of magnitudes, without any resonance in the upper layers.
- (2) Comment accepted.
- (3) Had been changed in the new manuscript.
- (1) Figure 1 is somewhat "overcrowded".
- (2) Comment accepted.
- (3) Had been changed in the new manuscript.
- (1) Page 2 lines 17-20 this should come earlier in this section.
- (2) Comment accepted.

- (3) Had been changed in the new manuscript.
- (1) Page 2 lines 26-27 several claims were made as to how is Vs30 suitable for Israel (i.e. Zaslavsky et al. 2012, Natural Science).
- (2) Although Zaslavsky et al. (2012) claimed that the use of Vs30 is a simplification that cannot be justified in the complex geological conditions as in Israel, no alternatives were proposed. Therefore, in this scenario, the Israel Standards Institute still adopts the Vs30 parameter.
- (3) We clarified this adding a statement to the manuscript.
- (1) Where is the Hough and Avni (2011) reference in the list?
- (2) Comment accepted.
- (3) Had been changed in the new manuscript.
- (1) Vs30 notation is not consistent throughout the manuscript.
- (2) Comment accepted.
- (3) Had been changed in the new manuscript.
- (1) Is the equation of Hough and Avni based on Bakun (2006) (page 3 line 28) or Bakun and Wentworth (1999) (page 5 line 2-3) or both, and where are these references in the list?
- (2) The equation is based on Bakun and Wentworth (1997).
- (3) References had been updated to the new manuscript.
- (1) Page 50 line 6 what do the authors mean by 60% prediction boundary? Is that the boundary for which 60% of the data is included within? Unclear.
- (2) Exactly
- (3) We redefined it in the new manuscript.
- (1) I'm not sure that section 4.1-4.2 belong in the discussion section. They seem to be more suitable in the methods section, since they are part of the modeling method.
- (2) We agree about that.
- (3) We moved this section to the proper section in the new manuscript.
- (1) Page 5 lines 17-18 what kind of data from the GII was yours compared to? This is very unclear. Is it borehole data? Refraction? HVSR? Is there a reference to the data?
- (2) The GII report (Aksinenko and Hofstetter, 2012) present all available geological, topographical, geophysical, geotechnical and borehole information to identify site-specific characteristics. This includes refraction and borehole.

- (1) Figure 10 -for which site are these results?
- (2) This figure refers to Beit Alfa site.
- (3) We clarified it in the new manuscript.
- (1) Page 7 line 3 Hough's name is spelled wrong. Also the citing is not complete.
- (2) Comment accepted.
- (3) Had been corrected in the new manuscript.
- (1) Page 7 line 9 Boore's equation is equation (4) and not (2). Also citing not complete.
- (2) Comment accepted.
- (3) Had been corrected in the new manuscript.
- (1) Figure 11 the legend is not clear does the curve represent the new GMPE, with the site term? Also I believe there is no need to use different symbols for amplified and de-amplified, they can be put in the same category as "MMI before site correction" or similar. Same goes for figure 7 there is no need to use different colors for the symbols, and the use of the term "amplified" or "de-amplified" is not accurate amplification is relative to reference rock conditions, and here I believe you mean the sites were "amplified" comparing to the GMPE.
- (2) Comment accepted.
- (3) Had been changed in the new manuscript.
- (1) Page 7 line 15 again Boore's equation is #4.
- (2) Comment accepted.
- (3) Had been corrected in the new manuscript.
- (1) It would seem to be useful if the authors use index numbers to identify their measurement locations in the figures, as they are numbered in table 1.
- (2) We thought over this but unfortunately, this leads to "overcrowded" figures.
- (1) In table 1 include the epicentral distance of the measurements. This will allow the reader to understand the statements they make regarding the misfit of the different locations.
- (2) Comment accepted.
- (3) Had been added in the new manuscript.

**Reviewer #2**

We thank the reviewer for pointing out many importing issues. In order to be clear, the response structure is in the prescribed sequence:

- (1) Comments from Referee
- (2) Author's response
- (3) Author's changes in the manuscript
- (1) Data are not shown in Chapter 3, and also the derivation of results and uncertainties are not given. This needs to be provided to the reader. As given here, one is not convinced.
- (2) Comment accepted.
- (3) We add a whole part of supplementary material which include all the MASW reports.
- (1) The optimization of the constant (p.7, line 6) is only mentioned but not explained.
- (2) C4 is optimized by the Least Squares Fitting (LSF) A mathematical procedure for finding the best-fitting curve to a given set of points by minimizing the sum of the squares of the offsets ("the residuals") of the points from the curve.
- (3) We added a brief explanation to the new manuscript.
- (1) A discussion of the directivity of amplification in entirely missing.
- (2) The data and analyses for 1927 are rudimentary, and we feel that for such a moderate event (M6.2) it is premature to include directivity.
- (3) We mention directivity it in the new manuscript.
- (1) General language: in parts very colloquial and not scientific/precise enough. Phrasing like somewhat, incomplete referencing, mentioning of authors without year of publication etc. might give an impression of rushed writing, that must be thoroughly revised.
- (2) We accept this comment.
- (3) Care is taken to avoid colloquial language and to properly reference sources in new manuscript.
- (1) Abstract: be more precise, what are specific results? This is rather an intro plus technical description.
- (2) We thank the reviewer for pointing this out.
- (3) Had been changed in the new manuscript.

- (1) Introduction: description of formula 1 is incomplete in text.
- (2) We thought the description of the formula is sufficient and we give the classical reference of Aki and Richards, 2002. Yet, we accept the comment.
- (3) We added an explanation on the actual behavior of the amplification as reflected on the formula.
- (1) Methods: it is not clear if the term offset is used properly. It rather seems to mean profile distance in meters?
- (2) In surface seismic acquisition, the horizontal distance from source to the first geophone defined as offset as we mentioned in our manuscript.
- (1) Results: this chapter is completely inappropriate, because no data is introduced, nothing described, and no workflow given to the reader. Thereby, it is not possible to judge quality of results and be confidential in the outcome.
- (2) A simple workflow is given in the method section. Regarding the data see item (3)
- (3) Data will be accessible at the supplementary material within the new manuscript.
- (1) Discussion: The remaining discussion should also include a discussion of comparing achieved results here with other authors methods and workflows, not for the region alone but especially seen in worldwide literature.
- (2) We consider our work as pioneering one with no parallels elsewhere. Yet we compared several sites from the GII's report (Aksinenko and Hofstetter, 2012) and had other relevant data been accessible for us we would certainly expand the comparison.
- (3) The first passage of the revised discussion addresses the reviewer's concern. In addition clarify in the new manuscript some issues in this regard (see also response to comment concerning Figure 9).
- (1) Most of the figures are only mentioned in the text but need proper description there, too.
- (2) We accept this comment.
- (3) Had been corrected in the new manuscript.
- (1) Figure 1: could be less scetchy.
- (2) We accept this comment.
- (3) Had been changed in the new manuscript.

- (1) Figure 2: map made by JKH -> give proper reference or include him as author, if considerable work was done.
- (2) We accept this comment.
- (3) Had been corrected in the new manuscript.
- (1) Figure 8+9: correct legend spelling.
- (2) Thanks for identifying this.
- (3) Had been corrected in the new manuscript.
- (1) Figure 9: what is second measurement? This is unexplained and not understandable.
- (2) Comment accepted.
- (3) We redefined it in the new manuscript.
- (1) Figure 11: incomplete references in legend, caption: dots mark suspected.....
- (2) Comment accepted.
- (3) Had been corrected in the new manuscript.

**Calibrating a New Attenuation Curve for the Dead Sea Region Using Surface Wave Dispersion Surveys in Sites Damaged by the 1927 Jericho Earthquake**

Darvasi Yaniv1, Agnon Amotz1

10

15

20

5 1The Fredy & Nadine Herrmann Institute of Earth Sciences, The Hebrew University of Jerusalem, 9190401, Israel Correspondence to: Darvasi Yaniv (yaniv.darvasi@mail.huji.ac.il)

Abstract. Strong motion data is not common around the Dead Sea region. Therefore, calibrating a new attenuation equation is a considerable challenge. However, the Holy Land has a remarkable historical archive, attesting to numerous regional and local earthquakes. Combining the historical record with modern-new seismic measurements will enhance the regional equation.

On 11 July 1927, a crustal rupture generated a moderate  $6.25M_L$  earthquake around the northern part of the Dead Sea. Up to five hundred people were killed and extensive destruction was recorded, even at places as far as 150 kilometers from the focus. We consider local near-surface properties, in particular, the shear-wave velocity, as an amplification factor. Where the shear-wave velocity is low, the seismic intensity at places far from the focus would likely be greater than expected from a standard attenuation curve. In this work, we used the Multi-Multichannel Analysis of Surface Waves (MASW) method to estimate seismic wave velocity at anomalous sites in Israel in order to calibrate a new attenuation equation for the Dead Sea region, based on 1927 macroseismic data integrated with modern measurements.

Our new attenuation equation contains a term which quantifies solely only lithological effects, whilst while factors such as building quality, foundation depth, topography, earthquake directivity, type of fault, etc., remained out of the equationour scope. Nonetheless, about 60% of the measured anomalous sites fit expectations; therefore, this new GMPE is statistically better than old ones and better fitting is achieved compared to other relevant attenuation equations.

From a local point of view, this is the first time that an integration between of historical data and modern shear-wave velocity profileseismic measurements improves the attenuation relation for the Dead Sea region. In the wider context, regions of low-to-moderate seismicity should use macroseismic historical earthquake data, together with modern measurements, in order to better estimate the peak ground acceleration or the seismic intensities caused by future earthquakes. This integration will conceivably lead to a better mitigation of damage from understanding of future earthquakes and improve maps of seismic hazardrisk.

**1** Introduction**

5

Generating a modern and applicable attenuation equation is one of applied seismologists main interests. Considering the Dead Sea area, for which instrumental strong motion data are not available, this task is particularly challenging. Using the Holy Land's historically rich database, researchers had defined seismic intensities and estimated earthquake locations. Investigating anomalous sites, with seismic intensities higher or lower than predicted from the basic regional attenuation

relation, may lead to a better attenuation equation. The local geological conditions can strongly influence the nature and severity of shaking at a given site. Assessing the local geological conditions by geophysical techniques at these anomalous sites, and adding a logarithmic term to a basic attenuation equation, should improve the equation.

 This work focuses on the 1927 event, but it is part of wider research which extends to additional earthquakes. The 1927

 10
 event was chosen as it is the only devastating one recorded during the instrumented period.

Our main goal in this research is a tighter constraint on the attenuation equation derived from this event. This should allowus to examine whether this preliminary work coincides with our expectations of site amplification and de-amplification due to the lithology.

**1.1 Site Response**

15 Ground motion is controlled by a number of variables, including source characteristics, source distance, propagation directivity, near-surface geology, etc. Elastic-The elastic properties of near-surface materials, and their effects on seismic wave propagation, are crucial for to earthquake and civil engineering, and in environmental and earth science studies.

Seismic surface waves are initiated generated at the moment that an-the earthquake wave front impinges on the surfaceoccurs. These waves spread out, and the surface shakes are traveling and shaking the surface as they pass. SurfaceThe

- 20 wave amplitude at the surface is controlledaffected by the mechanical properties of the rocks below. These rocks-often consist of low velocity weathered rock over bedrock with\_a\_much higher seismic velocityvelocities. When seismic waves pass from a high-velocity layer to a low-velocity layer, their amplitudes and duration typicallycan increase. The phenomena of site amplification, as a result of soft sediments overlying hard bedrock, is well known since the early days of seismology (Milne, 1898). Site-effects are also well known and were investigated after several major earthquakes: Mexico City 1985
- 25 (Singh et al., 1988), Yerevan 1989 (Borcherdt et al., 1989), San Francisco 1989 (Hough et al., 1990), Los Angeles 1994 (Hall et al., 1994) and Kobe, 1995 (Aguirre and Irikura, 1997). Therefore, local lithology is a crucial factor for estimating site amplification, defined as the amplitude ratio between the surface layer and the underlying bedrock. Site amplification at a specific site can be attributed to many factors, such as basin effects, focusing effects, topography, and reverberation of the seismic waves in the upper layers due to acoustic impedance differences (Figure 1).

2

Therefore, lithology is a crucial factor for estimating site amplification, defined as the amplitude ratio between the surface layer and the underlying bedrock. Site amplification is due reverberation of the seismic waves in the upper layers according to acoustic impedance differences (Figure 1). The amplification, *A*, is proportional to the reciprocal square root of the product of the shear-wave velocity, *Vs*. (Eq. (1)) (Aki and Richards, 2002):

5
$$A \propto \frac{1}{\sqrt{V_{\text{s}}...}}$$
,

(1)

where is the density of the investigated soil. As shear-wave velocity decreases by a given fraction the amplification increases by half that fraction (for a constant density). Since density plays a minor role (Moro, 2015; Xia et al., 1999) the  $V_s$  value can be used to represent site conditions.

The phenomena of site amplification as a result of soft sediments on top of hard bedrock is well known since the early days of seismology (Milne, 1898). Also, site-effects are well known and were investigated based on several major earthquakes: Mexico City 1985 (Singh et al., 1988), Armenia 1989 (Borcherdt et al., 1989), San Francisco 1989 (Hough et al., 1990), Los Angeles 1994 (Hall et al., 1994) and Kobe (Japan) 1995 (Aguirre and Irikura, 1997).

The most widely used index parameter in the classification of the soil response is the average shear-wave velocity in the uppermost 30 meters of sediment, the  $Vs_{30}$ . This index parameter was accepted for site classification in the USA – National

15 Earthquake Hazards Reduction Program – NEHRP(National Earthquake Hazards Reduction Program – NEHRP), (Building Seismic Safety Council, 2001), also iIn Europe by the new provisions of Eurocode 8 (BSI, 2011), and in Israel it is accepted by the design provisions for earthquake resistance of structures - SI 413 (The Standards Institution of Israel, 2013). Standards Institute (The Standards Institution of Israel, 2013). The value of 30 meters comes from the USA and European building codes, where it was found empirically that this depth is directly proportional to

20

10

comes nom the OSX and European building codes, where it was round empirically that this depth is directly proportional to deeper and shallower values (Boore et al., 2011). Zaslavsky et al. (2012) argued that the use of  $V_{320}$  is a simplification that cannot be justified in the complex geological conditions in Israel, yet no alternatives have thus far been proposed. Therefore, in this scenario, the Israel Standards Institute still adopts the  $V_{320}$  index. In Israel, there is not much data for this kind of correlation. Therefore, in this scenario, the Israel Standards Institute adopts the  $V_{320}$  parameter.

25

In modern attenuation equations, also known as ground motion prediction equations (GMPE), coefficients are derived set from strong motion data, namely ground acceleration measurements. In the past, and in areas lacking the technology to record earthquakes, it is impossible to measure the peak ground acceleration (PGA) directly directly measure the peak ground acceleration (PGA). Therefore, it is common to categorize historical earthquakes with seismic intensity scales that describe the damage at each site or area (Ambraseys, 2009; Guidoboni and Comastri, 2005).

3

**1.2 The M6.2 1927 Jericho 1927 earthquake**

5

The left-lateral Dead Sea transform separates the Sinai-Levant Block from the Arabian Plate (Figure 2). The 6.2ML July 11, 1927 Jericho earthquake (Ben-Menahem et al., 1976; Shapira, 1979) was the strongest and most destructive earthquake to hit the Holy Land during the past and current centuriescentury. Furthermore, it was the first to be instrumentally recorded by seismographs. Furthermore, for the first time an earthquake with epicenter in the Holy Land was recorded by seismographs.

[revised manuscript text omitted]

Rayleigh wave dispersion curves are obtained by the MASW module of the RadExPro® software, whose calculation procedure is based on a paper by Park et al. (1998), and also by the WinMASW® software. From all the dispersion images that we calculated from each offset shot (Figure 6), we choose the smoothest and clearest one (Figure 6) to compute the site's  $Vs_{30}$  profile. An inversion process then finds the shear-wave velocity profile whose theoretical dispersion curve is as close as possible to the experimental curve (Figure 6). The data and coefficients are automatically inverted via genetic algorithms

5

which represent an optimization procedure belonging to the classification of global-search methods. Genetic algorithms are commonly used to generate high-quality solutions to optimization and search problems by relying on bio-inspired operators such as mutation, crossover and selection Compared to traditional linear inversion methods based on gradient methods (Jacobian matrix) these inversion techniques produce a very reliable result in terms of precision and completeness (Moro et al., 2007).

Rayleigh wave dispersion curves were obtained by the MASW module of the RadExPro software in which the calculation procedure is based on a paper by Park et al. (1998). From all the dispersion curves that we picked for each site (Figure 6A), we chose the smoothest and clearest dispersion image (Figure 6B) to compute the site's Vs30 profile (Figure 6D). An inversion process then finds that shear wave velocity profile whose theoretical dispersion curve is as close as possible to the experimental curve (Figure 6C). This procedure is done by Occam's inversion which is part of the MASW module of the RadExPro software. During this process, the Root Mean Square (RMS) error between the curves is minimized while

2.2 Velocity model

5

10

All models were considered to be a stack of homogeneous linear elastic layers, neglecting lateral variations in soil properties. The number of unknowns for a layered model, when considering only shear-wave velocity, is three for each layer: density, thickness, and one elastic constant. Therefore, the number of unknowns is 3*n*-1 (where *n* represents the number of layers). The change in density with depth is usually small in comparison to the change in shear modulus and is normally neglected (Park et al., 1997).

**2.3 Number of layers & layer thicknesses**

maintaining the maximum model smoothness (Constable et al., 1987).

- 20 The resolution of surface wave surveys decreases with depth. Thin layers are well resolved when they are close to the surface, whereas at great depth, the resolution is limited and only large changes can be detected (Foti et al., 2014). Regardless of the number of the layers of the site, *Vs30* is almost the same in each case (Error! Reference source not found.Figure 7). For those reasons, as well as the lack of density information, we did not restrict each model to a specific number of layers. Without boreholes and lithostratigraphic data, which is the case in our work, a useful rule of thumb is to
- 25 assume layer thicknesses increasing with depth, to compensate for the decreased resolution with depth, an intrinsic shortcoming of surface wave testing (Foti et al., 2014).

**2.4 Depth of investigation**

We used a five-kilogram sledgehammer and summed up five strikes. For some sites, this type of source is insufficient to determinate a shear-wave profile down to 30 meters. In addition, at some sites, we were not able to spread the geophones at intervals of more than two meters, which limited the length of the seismic line. This length probably excludes longer

wavelengths which limits the depth of investigation. Lastly, as the shear-wave velocity of the lowest frequency is higher more data is available for deeper layers. Therefore, the penetration depth will decrease in areas with low shear-wave velocity. For instance, if we can clearly detect a phase velocity of about 300 m/sec at 5 Hz, we can roughly estimate a depth of investigation of approximately 20-30 meters according to the following equation:

$$= \frac{\left(\frac{Velocity_{f_{\min}}}{f_{\min}}\right)}{\left(\frac{f_{\min}}{f_{\min}}\right)}$$

п where n equals 2-3 (Foti et al., 2014; Moro, 2015). In other words, this equation emphasizes that the depth of investigation is about a half to a third of the largest wavelength observed.

**3 Results**

Ζ 5

Zohar and Marco (2012) relocated the epicenter to a point near the Almog settlement. We used this most recently published 10 epicenter for calculating the new epicentral distances, d. Figure 7Figure 8 shows a scatter plot of MMI vs. d for their 133 sites. Hough and Avni (2011) fit this data with a curve which best describes the attenuation relation for this event. Using the mathemathical form of their curve, we calculated upper and lower limits such that 60% of the points are enclosed. This we call the 60% prediction boundary. We consider that the lithological effects probably account for much of the scatter beyond this boundry, due to amplification and de-amplification.

Avni's (1999) original attenuation equation yields a R2 of 0.26. Hough & Avni's (2011) revised equation, based on Bakun 15 and Wentworth (1997), yields the same fit. Identifying the Kalia fault as the source location (Kagan et al., 2011) in Eq. (2) yields a fit of 0.35. The best fit of 0.38 is obtained using the Almog settlement as the epicenter (Zohar and Marco, 2012). Accordingly, our analysis is based on this epicenter location. A scatter diagram of the distribution of all 133 sites for which Avni (1999) estimated seismic intensity, together with a prediction boundary of 60% from Eq.(2), highlight the sites that 20 were amplified and de-amplified (Figure 7).

**3.1 MASW surveys**

From these 24 surveys, we succeeded in extracting Vs30 for 19 of 20 sites (the Hartuv data were too noisy for interpretation) (Table 2).

Field Code Changed

(3)

25

**4 Discussion**

5

10

A number of researchers have studied the 1927 event. Avni (1999) tried to reduce the impact of local geology and attempted to generate basic attenuation curves for specific azimuths. Zohar and Marco (2012) relocated the source position while Shani-Kadmie et al. (2016) studied directivity of the source pattern. None of these publications address the  $Vs_{30}$ measurements. An attenuation equation with a term that depends on the  $Vs_{30}$  index should lead to a better understanding of past events, and to more useful predictions of future earthquakes.

**4.1 Survey locations and validation**

The decision as to where exactly each survey should would take place was based on Avni's thesis (Avni, 1999). WhereIf the location was not sufficiently clear-known, we rechecked the reference given by Avni. In most cases, there was evidence of specific damaged buildings. We tried to locate those-these buildings while looking aton historical maps (1927-1945). Unfortunately, most sites were located inside urban areas, where we could not execute-carry out the seismic surveys. Therefore, we surveyed in nearby open areas as near-close as possible to the referenced damage zones.

To validate our results, we compared them with a summary of thousands of seismic evaluations around Israel carried out over the years by the Geophysical Institute of Israel (GII), and compiled in a report by (Aksinenko and Hofstetter, 2012).

These evaluations were based upon refraction and borehole velocity measurements yielding *Vs* and/or *Vp* values, as well as the effects of topography and geology. The spacing of their data was such that often a number of GII sites had to be averaged to provide a value within several kilometers for comparison with our MASW values. However, Figure 9 shows that the GII-based values are in consistent agreement with those of the MASW. However, this comparison is a bit tricky because *Vs*30 results for two sites 5 km or much less distant could be significantly different, as shown in Figure 9Figure 10. Remembering that *Vs*30 enters a logarithmic term, we find our approach potentially useful.

To validate our results, we compared them with the nearest (up to three kilometers distance) data from The Geophysical Institute of Israel (GII) and a reasonable fit was achieved (Figure 8). However, this comparison is a bit tricky because Vs30 results for two sites distant three kilometers or much less could be significantly different, as shown in Figure 9. Remembering that Vs30 enters a logarithmic term, we find our approach potentially useful.

**25 4.2 A new attenuation equation**

In the present case of the 1927 earthquake, the sources of the data are mostly historical documents and not strong data measurements. This makes it difficult to quantify site response into a single equation. In the practical modern attenuation relationship,  $Vs_{30}$  is a crucial index. A term that depends on  $Vs_{30}$  has previously been constrained for several large data sets (Abrahamson et al., 2014; Boore et al., 1997; Campbell and Bozorgnia, 2008). We chose the Boore et al. (1997) attenuation

30 equation (Eq. (4)) in order to emphasize site response.

 $\ln Y = b_1 + b_2(M - 6) + b_3(M - 6)^2 + b_5 \ln(r) + b_v \ln\left(\frac{Vs}{V_A}\right)$

where Y is the ground-motion variable (peak horizontal acceleration or pseudo-acceleration response in g), M is the moment magnitude, r is the epicentral distance in kilometers,  $V_{A_a}$  and all b terms are frequency dependent coefficients to be determined. By adding Boore et al.'s (1997) Vs term to Hough and Avni (2011) attenuation equation (Eq. (2)), we suggest a new equation for the region:

$$MMI = -0.64 + 1.7M - 0.00448d - 1.67\log(d) + C_4 \ln\left(\frac{Vs_{30}}{V_A}\right)$$

where  $V_A$  and  $C_d$  are adjustable coefficients. The first four coefficients remain the same as we assert that the magnitude, attenuation, geometrical spreading and site response are all independent. We adopt the value of  $V_A$  from Boore's (1987) equation (Eq. (4)), as it represents a single value independent of the frequency. We took formerly derived GMPE, with its coefficients, and added another term, by regressing only for the new coefficient, then optimizing  $C_d$  and  $V_A$  by Least Squares Fitting (LSF), as shown in **Error! Reference source not found.**Figure 11 we get the final equation:

$$MMI = -0.64 + 1.7M - 0.00448d - 1.67\log(d) - 2.1\ln\left(\frac{Vs_{30}}{655}\right)$$

**4.2 Velocity model**

5

10

25

All models were considered as a stack of homogeneous linear elastic layers, neglecting lateral variations in soil properties. The number of unknowns for a layered model, when considering only shear wave velocity, is three for each layer: density, thickness, and one elastic constant. Therefore, the number of unknowns is 3n-1 (where n represents the number of layers). The change in density with depth is usually small in comparison to the change in shear modulus and is normally neglected (Park et al., 1997). Therefore we set the density to 2000 [kg/m3] for all layers in all our sites and thus the number of unknowns decreases to 2n-1.

**20 4.2.1 Number of layers & layer thicknesses**

The resolution of surface wave surveys decreases with depth. Thin layers are well resolved when they are close to the surface, whereas at great depth, the resolution is limited and only large changes can be detected. The reduction of the sensitivity with depth results in a loss of resolution, or in the ability to identify the properties of thin layers. Thus, these features cannot be accurately resolved (Foti et al., 2014). Regardless of the number of the layers of the site,  $Vs_{30}$  is almost the same in each case (Figure 10). For those reasons, as well as the lack of density information, we did not restrict each model to a specific number of layers. Without boreholes and lithostratigraphic data, which is the case in our work, a good

9

Field Code Changed

(4)

(5)

(6)

Field Code Changed

Field Code Changed

rule of thumb is to assume layer thicknesses increasing with depth, to compensate for the decreased resolution with depth, which is an intrinsic shortcoming of surface wave testing (Foti et al., 2014).

**4.2.2 Depth of investigation**

5

10

We used a five-kilogram sledgehammer and summed up five strikes. In some sites this type of source is insufficient to determinate a shear-wave profile down to 30 meters. In addition, at some sites we were not able to spread the geophones at intervals of more than two meters which limited the length of the seismic line. This length probably excludes longer wavelengths which limits the depth of investigation. Lastly, as the shear-wave velocity of the lowest frequency is higher – more data is available for deeper layers. Therefore, the penetration depth will decrease in areas with low shear-wave velocity. For instance, if we can clearly detect a phase velocity of about 200-300 m/sec at 5 Hz, we can roughly estimate a depth of investigation of approximately 12-20 meters according to the following equation:

$$\frac{\frac{\left(\frac{Velocity_{f_{\min}}}{f_{\min}}\right)}{Z = \frac{n}{n}},$$

where *n* equals 2-3 (Foti et al., 2014; Moro, 2015). In other words, this equation emphasizes that the depth of investigation is about a half to a third of the largest wavelength observed.

(3)

**4.3 The performance of the new attenuation equation**

- With these coefficients, 58% or 11 of 19 sites, were amplified or de-amplified as we expected. For the entire distance range (up to 250 km) the Vs30 corrections leave 42% sites out of the prediction boundary (eight of nineteen sites). Seismic intensities at all these eight sites are overpredicted by the attenuation equation (Eq. (2)) (Figure 11Figure 12). We expect that Vs30 at these sites will be higher than 655 m/sec in order to obtain de-amplification. However, our results show the opposite effect these eight sites are characterized by lower Vs30 which drives amplification. This can be caused by the fact that measurements were taken over agricultural fields, of which the upper layers (the first few meters) are characterized by low
- shear-wave velocity, decreasing the average Vs. Another reasonable explanation is that we did not succeed in extracting the average shear-wave velocity down to 30 meters and perhaps we missed some high-velocity shear-wave layers at deeper layers. In such cases, we constrain the last layer to be thicker in order to estimate  $Vs_{30}$  for all our surveys.

**4.3 A new attenuation equation**

25 In the present case of the 1927 earthquake, the sources of the data are mostly historical documents and not measurements. This makes it difficult to quantify site response into a single equation. In the practical modern attenuation relation,  $V_{s_{30}}$  is a crucial index. A term that depends on  $V_{s_{30}}$  has been constrained for several large data sets (Abrahamson et al., 2014; Boore et al., 1997; Campbell and Bozorgnia, 2008). We chose the Boore et al. (1997) attenuation equation (Eq. (4)) in order to emphasize site response.

$$\ln Y = b_1 + b_2 (M - 6) + b_3 (M - 6)^2 + b_5 \ln(\mathbf{r}) + b_\nu \ln\left(\frac{Vs}{V_A}\right), \tag{4}$$

where *Y* is the ground-motion variable (peak horizontal acceleration or pseudo-acceleration response in g), *M* is the moment magnitude, *r* is the epicentral distance in kilometers,  $V_{A_r}$  and all *b* are frequency dependent coefficients to be determined. By adding Boore et al.'s Vs term to Hoguh and Avni's attenuation equation (Eq. (2)), we suggest a new equation for the Dead Sca region:

5

10

15

20

$$MMI = -0.64 + 1.7M - 0.00448d - 1.67\log(d) + C_4 \ln\left(\frac{Vs_{30}}{Vs_{refernce}}\right),$$
(5)

where  $Vs_{referance}$  is the shear-wave velocity of the bedrock and  $C_{4}$  is an adjustable constant. Optimizing this constant to our data yields the final equation:

 $MMI = -0.64 + 1.7M - 0.00448d - 1.67\log(d) - 1.8\ln\left(\frac{Vs_{30}}{760}\right),\tag{6}$

We adopt here the value of  $V_A$  from Boore's equation (equation 2) as representing  $Vs_{referrce}$ . According to most national standards, including the one in Israel (SI #413), the reference bedrock shear wave velocity is set equal to 760 m/sec for all frequencies in the entire region. With these coefficients, 58% or 11 of 19 sites, were amplified or de-amplified as we expected. Based on the new attenuation equation (Eq. (6)) we reduced the site-effects (Figure 11) and compare the fitness of our attenuation curve with those of Avni (1999) and Hough & Avni (2011) (Figure 12). The new attenuation equation fits the data somewhat better for all four epicenters.

Boore et al.'s equation (Eq. (2)) is restricted for use only for earthquakes of magnitude 5.5-7.5 and epicentral distance up to 80 kilometers. After lithological corrections, sites located up to 70 kilometers from the epicenter are well predicted: the entire population of six anomalous sites shifted to the prediction boundary. On the other hand, sites located farther than 70 kilometers from the epicenter converge into the prediction boundary to a lesser extent (five sites of thirteen which are 38%) (Figure 11). This observation is consistent with Boore's restriction.

For the entire distance range (up to 250 km) the Vs30 corrections leave 42% sites out of the prediction boundary (eight of nineteen sites). Seismic intensities in all these eight sites are underpredicted by the attenuation equation (Eq. (2)) (Figure 11).
 We expect that Vs30 at these sites will be higher than 760 m/sec in order to obtain de-amplification. However, our results show the opposite effect – these eight sites are characterized by lower Vs30 which drive amplification. This can be caused by the fact that measurements were taken over agricultural fields, of which the upper layers (the first few meters) are

characterized by low shear-wave velocity, decreasing the average Vs. Another reasonable explanation is that we did not succeed in extracting the average shear-wave velocity down to 30 meters and perhaps we missed some high velocity shear-wave layers at deeper layers. In such cases, we constrain the last layer to be thicker in order to estimate Vs30 for all our surveys.

**5 5 Conclusions**

In this research, we investigated site amplification and de-amplification around Israel. According to previous studies (Aki, 1988; Boore, 2003; Borcherdt, 1994; Field and Jacob, 1995; Joyner and Boore, 1988) the local lithology can amplify or deamplify wave amplitude. The commonly used modern seismic method – MASW – allows the extraction of  $\sqrt{s}$  profiles at 20 19 sites reportedly damaged by the 1927 6.2ML6.2 earthquake. We use these profiles to update the attenuation equation for the Dead Sea region by including the  $\sqrt{s_{30}}$  term.

10

According to this new equation, 11 sites, which constitute 58% of our measured samples, move into the 60% prediction boundary. This suggests that the prediction boundary actually encompasses over 80% of the macroseismic observations. This fit is better than any available attenuation equation for the Dead Sea region-(Figure 11). However, as we have used only 19 sites, we should consider further research and provide wider results. Although our final equation (Eq. (6)) shows

- 15 amplification and de-amplification depending on  $Vs_{30\pm}$  it does not take into consideration any other factor such as building quality, foundation depth, topography, earthquake directivity, type of fault etc. Obviously, for better results we must use more methods and jointly invert some other seismic data such as: refraction (S and P waves), Horizontal to Vertical Spectral Ratio (HVSR), MASW of the transverse component of Love waves, MASW of the radial component of Rayleigh wave, Extended Spatial Auto-Correlation (ESAC), etc. Also, with these data in hand, a full inversion for the epicenter will be in
- 20 order.

25

Despite the scarcity of data, this is the first time that an integration of historical data with shear-wave velocity profile modern seismic-measurements improves the attenuation relation. In order to better estimate the peak ground acceleration or the seismic intensities that will caused by future earthquakes, attenuation relations are necessary for areas characterized by high seismicity. worldwide, especially in areas characterized by high seismicity. Some of the regions of low to moderate seismicity have rich sources of historical earthquake data. The integration of historical data with shear-wave velocity profilesmodern measurements will lead to a better understanding of future earthquakes.

Acknowledgments. We thank the Neev Center for Geoinfomatics's's facility and its students. We are especially grateful to Dr. John K. Hall, who-founderd of the Center, for his ongoing support. We are grateful to the Helmholtz Association of German Research Centers for funding this research. We thank Prof. Moshe Reshef for comments and suggestions on an earlier draft and Prof. Ran Bachrach for valuable advice. We acknowledge the contribution of Prof. Michael Weber and the geophysical deep sounding section at GFZ. Finally, we thank Amit Ronen for his assistance.

12

| 5  | References                                                                                                                       |
|----|----------------------------------------------------------------------------------------------------------------------------------|
|    | Abel, F. M.: No Le Recent Tremblement de Terre en Palestine, Rev. Biblique, 36, 571–578, 1927.                                   |
|    | Abrahamson, N. A., Silva, W. J. and Kamai, R.: Summary of the ASK14 ground motion relation for active crustal regions.           |
|    | Earthq. Spectra, 30(3), 1025–1055, 2014.                                                                                         |
|    | Aguirre, J. and Irikura, K.: Nonlinearity, liquefaction, and velocity variation of soft soil layers in Port Island, Kobe, during |
| 10 | the Hyogo-ken Nanbu earthquake, Bull. Seismol. Soc. Am., 87(5), 1244–1258, doi:10.1144/pygs.51.3.177, 1997.                      |
|    | Aki, K.: Local site effects on ground motion, Earthq. Eng. Soil Dyn. II-Recent Adv. Gr. Motion Eval. Geotech. Spec.              |
|    | Pudlication, 20, 103–155, 1988.                                                                                                  |
|    | Aki, K. and Richards, P. G.: Quantitative seismology, University Science Books., 2002.                                           |
|    | Aksinenko, T. and Hofstetter, A.: 1-D semi-empricical modeling of the subsurface across Israel for site effect evaluations.,     |
| 15 | 2012.                                                                                                                     |
|    | Ambraseys, N.: Earthquakes in the Mediterranean and Middle East: a multidisciplinary study of seismicity up to 1900,             |
|    | Cambridge University Press., 2009.                                                                                               |
|    | Ambraseys, N. N. and Melville, C. P.: An analysis of the eastern Mediterranean earthquake of 20 May 1202, Hist. Seism.           |
|    | earthquakes world, 181–200, 1988.                                                                                                |
| 20 | Amiran, D. H: A Revised Earthquake-Catalouge of Palastine, Isr. Explor. J., 2, 48-65, 1951.                                      |
|    | Arieh, A.: Seismicity of Israel and Adjacent Area, Minist. Dev., 43, 1–14, 1967.                                                 |
|    | Avni, R.: The 1927 Jericho Earthquake. Comprehensive Macroseismic Analysis Based on Contemporary Sources., Ben                   |
|    | Gurion University of the Negev, Beer Sheva (in Hebrew)., 1999.                                                                   |
|    | Bakun, W. H. and Wentworth, C. M.: Estimating earthquake location and magnitude from seismic intensity data, Bull.               |
| 25 | Seismol. Soc. Am., 87(6), 1502–1521, 1997.                                                                                |
|    | Ben-Menahem, A.: Four thousand years of seismicity along the Dead Sea Rift, J. Geophys. Res., 96(B12), 20195,                    |
|    | doi:10.1029/91JB01936, 1991.                                                                                              |
|    | Ben-Menahem, A., Nur, A. and Moshe, V.: Tectonics, seismicity and structure of the Afro - Eurasian junction - the breaking       |
|    | of an incoherent plate, , 12(1), 1–50, 1976.                                                                              |
| 30 | Blankenhorn, M.: Das Erdbeben in Juli 1927 in Palestina, Zeitschr.D.Pal, (51), 123–125, 1927.                                    |
|    | Boore, D. M.: Simulation of ground motion using the stochastic method, Pure Appl. Geophys., 160(3), 635-676,                     |
|    | doi:10.1007/PL00012553, 2003.                                                                                             |
|    | Boore, D. M., Joyner, W. B. and Fumal, T. E.: Equations for Estimating Horizontal Response Spectra and Peak Acceleration         |
|    | from Western North American Earthquakes: A Summary of Recent Work, Seismol. Res. Lett., 68(1), 128-153,                          |

|    | doi:10.1785/gssrl.76.3.368, 1997.                                                                                             |  |  |  |  |
|----|-------------------------------------------------------------------------------------------------------------------------------|--|--|--|--|
|    | Boore, D. M., Thompson, E. M. and Cadet, H.: Regional correlations of Vs30 and velocities averaged over depths less than      |  |  |  |  |
|    | and greater than 30 meters, Bull. Seismol. Soc. Am., 101(6), 3046-3059, doi:10.1785/0120110071, 2011.                         |  |  |  |  |
|    | Borcherdt, R., Glassmoyer, G., Andrews, M. and Cranswick, E.: Effect of site conditions on ground motion and damage,          |  |  |  |  |
| 5  | Earthq. spectra, 5(S1), 23-42, 1989.                                                                                          |  |  |  |  |
|    | Borcherdt, R. D.: Estimates of site-dependent response spectra for design (methodology and justification), Earthq. spectra,   |  |  |  |  |
|    | 10(4), 617–653, 1994.                                                                                                  |  |  |  |  |
|    | Brawer, A. Y.: Earthquakes events in Israel from July 1927 to August 1928, 1928.                                              |  |  |  |  |
|    | BSI: Eurocode 8 : Design of structures for earthquake resistance, 3, 2011.                                                    |  |  |  |  |
| 10 | Building Seismic Safety Council: NEHRP Recomendations for Seismic Regulations for New Buildings and Other                     |  |  |  |  |
|    | Structures, Part 1 : Provisions (FEMA - 368), (Fema 368), 392, 2001.                                                          |  |  |  |  |
|    | Campbell, K. W. and Bozorgnia, Y.: NGA ground motion model for the geometric mean horizontal component of PGA,                |  |  |  |  |
|    | PGV, PGD and 5% damped linear elastic response spectra for periods ranging from 0.01 to 10 s, Earthq. Spectra, 24(1),         |  |  |  |  |
|    | 139–171, doi:10.1193/1.2857546, 2008.                                                                                         |  |  |  |  |
| 15 | Ciaccio, M. G. and Cultrera, G.: Terremoto e rischio sismico, Ediesse., 2014.                                                 |  |  |  |  |
|    | Field, E. H. and Jacob, K. H.: A comparison and test of various site-response estimation techniques, including three that are |  |  |  |  |
|    | not reference-site dependent, Bull. Seismol. Soc. Am., 85(4), 1127-1143, 1995.                                                |  |  |  |  |
|    | Foti, S., Lai, C., Rix, G. and Strobbia, C.: Surface Wave Methods for Near-Surface Site Characterization., 2014.              |  |  |  |  |
|    | Guidoboni, E. and Comastri, A.: Catalogue of Earthquakes and Tsunamis in the Mediterranean Area from the 11th to the          |  |  |  |  |
| 20 | 15th Century, Istituto nazionale di geofisica e vulcanologia., 2005.                                                          |  |  |  |  |
|    | Hall, J.: The 25-m DTM (Digital Terrain Model) of Israel, Isr. J. Earth Sci., 57(3-4), 145-147, doi:10.1560/IJES.57.3-4.145,  |  |  |  |  |
|    | 2008.                                                                                                                         |  |  |  |  |
|    | Hall, J. F., Holmes, W. T. and Somers, P.: Northridge earthquake, January 17, 1994, Prelim. Reconnaiss. Rep., 1994.           |  |  |  |  |
|    | Hough, S. E. and Avni, R.: The 1170 and 1202 CE Dead Sea Rift earthquakes and long-term magnitude distribution of the         |  |  |  |  |
| 25 | Dead Sea Fault Zone, Isr. J. Earth Sci., 58(3), 295–308, doi:10.1560/IJES.58.3-4.295, 2011.                                   |  |  |  |  |
|    | Hough, S. E., Friberg, P. A., Busby, R., Field, E. F., Jacob, K. H. and Borcherdt, R. D.: Sediment-induced amplification and  |  |  |  |  |
|    | the collapse of the Nimitz Freeway, Nature, 344(6269), 853-855, doi:10.1038/344853a0, 1990.                                   |  |  |  |  |
|    | Joyner, W. B. and Boore, D. M.: Measurement, characterization, and prediction of strong ground motion, 1988.                  |  |  |  |  |
|    | Kagan, E., Stein, M., Agnon, A. and Neumann, F.: Intrabasin paleoearthquake and quiescence correlation of the late            |  |  |  |  |
| 30 | Holocene Dead Sea, J. Geophys. Res. Solid Earth, 116(4), 1–27, doi:10.1029/2010JB007452, 2011.                                |  |  |  |  |
|    | Miller, R. D., Xia, J., Park, C. B., Survey, K. G., Hunter, J. A. and Harris, J. B.: Comparing Shear-Wave Velocity Profiles   |  |  |  |  |
|    | Inverted From Multi- Channel Surface Wave With Borehole Measurements, , 18, 181–190, 2002.                                    |  |  |  |  |
|    | Milne, J.: Seismology: London, Kegan Paul, Trench, Truber, 1898.                                                              |  |  |  |  |
|    | Moro, G. D.: Surface Wave Analysis for Near Surface Applications., 2015.                                                      |  |  |  |  |

Moro, G. D., Pipan, M. and Gabrielli, P.: Rayleigh wave dispersion curve inversion via genetic algorithms and Marginal Posterior Probability Density estimation, 61, 39–55, doi:10.1016/j.jappgeo.2006.04.002, 2007.

Park, C. B., Miller, R. D. and Xia, J.: Multi-Channel Analysis of Surface Waves (MASW) prepared by., 1997.

Park, C. B., Miller, R. D. and Xia, J.: Imaging dispersion curves of surface waves on multi-channel record, in SEG Technical Program Expanded Abstracts 1998, vol. 17, pp. 1377–1380, Society of Exploration Geophysicists., 1998.

Raphael, K. and Agnon, A.: Earthquakes East and West of the Dead Sea Transform in the Bronze and Iron Ages, 769–798 in Shai et al., 2018.

Ryden, N., Park, C. B., Ulriksen, P. and Miller, R. D.: Multimodal Approach to Seismic Pavement Testing, J. Geotech. Geoenvironmental Eng., 130(6), 636–645, doi:10.1061/(ASCE)1090-0241(2004)130:6(636), 2004.

10 Shani-Kadmie, S., Tsesarsky, M. and Gvirtzman, Z.: Distributed slip model for forward modeling strong Earthquakes, Bull. Seismol. Soc. Am., 106(1), 93–103, doi:10.1785/0120150102, 2016.

Shapira, A.: Redetermined magnitudes of earthquakes in the Afro-Eurasian Junction, Isr. J. Earth Sci, 28, 107–109, 1979.

Shapira, A., Avni, R. and Amos, N.: A new estimate for the epicenter of the Jericho earthquake of 11 July 1927, Isr. J. Earth-Sciences, 42(2), 93–96, 1993.

15 Singh, S. K., Lermo, J., Dominguez, T., Ordaz, M., Espinosa, J. M., Mena, E. and Quaas, R.: The Mexico earthquake of September 19, 1985-A study of amplification of seismic waves in the valley of Mexico with respect to a hill zone site, Earthq. spectra, 4(4), 653–673, 1988.

The Standards Institution of Israel: Design provisions for earthquake resistance of structures - SI 413, , (5), 2013. Willis, B.: Earthquakes in the Holy Land, Bull. Seismol. Soc. Am., 18(2), 73–103, 1928.

 Xia, J., Miller, R. D. and Park, C. B.: Estimation of near surface shear wave velocity by inversion of Rayleigh waves, Geophysics, 64(3), 691–700, doi:10.1190/1.1444578, 1999.
 Zaslavsky, Y.: Questioning the applicability of soil amplification factors as defined by NEHRP (USA) in the Israel building standards, Nat. Sci., 04(28), 631–639, doi:10.4236/ns.2012.428083, 2012.

Zohar, M. and Marco, S.: Re-estimating the epicenter of the 1927 Jericho earthquake using spatial distribution of intensity
 data, J. Appl. Geophys., 82, 19–29, doi:10.1016/j.jappgeo.2012.03.004, 2012.

Figure 1: Schematic view of site amplification due to reverberations. Seismogram at the surface shows amplification in comparison to the seismogram located over the bedrock. See new fig

---

## Author Response (AR2)

Dear Lotte Krawczyk,

Thank you very much for the comments.

As you suggested, we sorted again Chapters 2 and 3: In Chapter 2 we clarified only the method while in Chapter 3 we expanded the results together with real data example. Now, this version is more logical and fluent for reading.

Best Regards,

**Calibrating a New Attenuation Curve for the Dead Sea Region Using Surface Wave Dispersion Surveys in Sites Damaged by the 1927 Jericho Earthquake**

Darvasi Yaniv[1], Agnon Amotz[1]

[1]The Fredy & Nadine Herrmann Institute of Earth Sciences, The Hebrew University of Jerusalem, 9190401, Israel

*Correspondence to*: Darvasi Yaniv (yaniv.darvasi@mail.huji.ac.il)

**Abstract.** Instrumental sStrong motion data is not common around the Dead Sea region. Therefore, calibrating a new attenuation equation is a considerable challenge. However, the Holy Land has a remarkable historical archive, attesting to numerous regional and local earthquakes. Combining the historical record with new seismic measurements will enhance the regional equation.

On 11 July 1927, a rupture, in the crust in proximity to the Northern Dead Sea, generated a moderate 6.2$M_L$ earthquakeOn 11 July 1927, a crustal rupture generated a moderate 6.2$M_L$ earthquake around the northern part of the Dead Sea. Up to five hundred people were killed, and extensive destruction was recorded, even as far as 150 kilometers from the focus. We consider local near-surface properties, in particular, the shear-wave velocity, as an amplification factor. Where the shear-wave velocity is low, the seismic intensity far from the focus would likely be greater than expected from a standard attenuation curve. In this work, we used the Multichannel Analysis of Surface Waves (MASW) method to estimate seismic wave velocity at anomalous sites in Israel in order to calibrate a new attenuation equation for the Dead Sea region.

Our new attenuation equation contains a term which quantifies only lithological effects, while factors such as building quality, foundation depth, topography, earthquake directivity, type of fault etc., remain out of our scope. Nonetheless, about 60% of the measured anomalous sites fit expectations; therefore, this new GMPE is statistically better than old ones.

From a our local point of view, this is the first time that integration of the 1927 historical data and modern shear-wave velocity profile measurements improves the attenuation relationship equation (sometimes referred to as the attenuation relation) for the Dead Sea region. In the wider context, regions of low-to-moderate seismicity should use macroseismic earthquake data, together with modern measurements, in order to better estimate the peak ground acceleration or the seismic intensities caused by future earthquakes. This integration will conceivably lead to a better mitigation of damage from future earthquakes and improve maps of seismic hazard.

**1 Introduction**

Generating a modern and applicable attenuation equation is one of applied seismologists main interests. Considering the Dead Sea area, for which instrumental strong motion data are not available, this task is particularly challenging. Using the Holy Land's historically rich database, researchers had defined seismic intensities and estimated earthquake locations. Investigating anomalous sites, with seismic intensities higher or lower than predicted from the basic regional attenuation relation, may lead to a better attenuation equation. The local geological conditions can strongly influence the nature and severity of shaking at a given site. Assessing the local geological conditions by geophysical techniques at these anomalous sites, and adding a logarithmic term to a basic attenuation equation, should improve the equation.

This work focuses on the 1927 event, but it is part of wider research which extends to additional earthquakes. The 1927 event was chosen as it is the only devastating one recorded, albeit teleseismically, during the instrumented period.

Our main goal in this research is a tighter constraint on the attenuation equation derived from this event. This should allow us to examine whether this preliminary work coincides with our expectations of site amplification and de-amplification due to the local lithology.

**1.1 Site Response**

Ground motion is controlled by a number of variables, including source characteristics, source distance, propagation directivity, near-surface geology, etc. The elastic properties of near-surface materials, and their effect on seismic wave propagation, are crucial to earthquake and civil engineering, and environmental and earth science studies.

Seismic surface waves are initiated at the moment that the earthquake wave front impinges on the surface. These waves spread out, and the surface shakes as they pass. Surface wave amplitude at the surface is controlled by the mechanical properties of the rocks below. These often consist of low velocity weathered rock over bedrock with much higher velocities. When seismic waves pass from a high-velocity layer to a low-velocity layer, their amplitudes and duration typically increase. The phenomena of site amplification, as a result of soft sediments overlying hard bedrock, is well known since the early days of seismology (Milne, 1898). Site-effects are also well known and were investigated after several major earthquakes: Mexico City 1985 (Singh et al., 1988), Yerevan 1989 (Borcherdt et al., 1989), San Francisco 1989 (Hough et al., 1990), Los Angeles 1994 (Hall et al., 1994) and Kobe, 1995 (Aguirre and Irikura, 1997). Therefore, local lithology is a crucial factor for estimating site amplification, defined as the amplitude ratio between the surface layer and the underlying bedrock. Site amplification at a specific site can be attributed to many factors, such as basin effects, focusing effects, topography, and reverberation of the seismic waves in the upper layers due to acoustic impedance differences (Figure 1).

The amplification, $A$, is proportional to the reciprocal square root of the product of the shear-wave velocity, $Vs$. (Eq. (1)) (Aki and Richards, 2002):

$$A \propto \frac{1}{\sqrt{V_s \rho}} , \tag{1}$$

where $\rho$ is the density of the investigated soil. As shear-wave velocity decreases by a given fraction the amplification
5   increases by half that fraction (for a constant density). Since density plays a minor role (Dal Moro, 2014; Xia et al., 1999) the $V_s$ value can be used to represent site conditions.

The most widely used index in the classification of the soil response is the average shear-wave velocity in the uppermost 30 meters, the $Vs_{30}$. This index was accepted for site classification in the USA – National Earthquake Hazards Reduction Program – NEHRP (Building Seismic Safety Council, 2001). In Europe by the new provisions of Eurocode 8 (BSI, 2011),
10   and in Israel it is accepted by the design provisions for earthquake resistance of structures - SI 413 (The Standards Institution of Israel, 2013). The value of 30 meters comes from the USA and European building codes, where it was found empirically that this depth is directly proportional to deeper and shallower values (Boore et al., 2011). Zaslavsky et al. (2012) argued that the use of $Vs_{30}$ is a simplification that cannot be justified in the complex geological conditions in Israel, yet no alternatives have thus far been proposed. Therefore, in this scenario, the Israel Standards Institute still adopts the $Vs_{30}$ index.

15   In modern attenuation equations, also known as ground motion prediction equations (GMPE), coefficients are derived from strong motion data, namely from ground acceleration measurements. In the past, and in areas lacking the technology to record earthquakes, it was impossible to measure the peak ground acceleration (PGA) directly. Therefore, it is common to categorize historical earthquakes with seismic intensity scales that describe the damage at each site or area (Ambraseys, 2009; Guidoboni and Comastri, 2005)

20   **1.2 The M6.2 1927 Jericho earthquake**

The left-lateral Dead Sea transform separates the Sinai-Levant Block from the Arabian Plate (Figure 2). The $6.2M_L$ July 11, 1927, Jericho earthquake (Ben-Menahem et al., 1976; Shapira, 1979) was the strongest and most destructive earthquake to hit the Holy Land during that century. Furthermore, it was the first to be instrumentally recorded by seismographs. The epicentral location was originally estimated at a few kilometers south of the Damia Bridge, which is 30 kilometers north of
25   Jericho (International Seismological Summary – ISS Bulletin of 1927). In the following decades new estimates have been published: Shapira et al. (1993) calculated the epicenter to be near Mitzpe Shalem. Zohar and Marco (2012) estimated the epicenter to be near the Almog settlement, about 30 kilometers north of Shapira's epicenter, and Kagan et al. (2011), surmised that the source was somewhere on the Kalia fault, located in the northern part of the Dead Sea graben, perpendicular to the main Dead Sea fault (Figure 2).

The damage from the earthquake was heavy, especially in places near the source, but not only there: In Nablus, located 70 km from the epicenter (Figure 2), 60 people were killed, 474 were injured, and more than 700 structures were destroyed, most of which were built on soft sediments (Blankenhorn, 1927; Willis, 1928). By comparison, Jerusalem is only about 30 kilometers from the source and the damage there was much smaller, especially in property. However, in Mount Scopus and

5   the Mount of Olives (eastern neighborhoods in Jerusalem), the damage exceeded that in other parts of Jerusalem (Abel, 1927; Brawer, 1928). Other cities also suffered from this earthquake:  Tens of people were injured and even died, and hundreds of houses were ruined in Ramleh and Lod (Brawer, 1928). Jericho in the Jordan Valley also suffered significant damage, especially in terms of buildings collapsing (Figure 3). The total number of victims was about 350-500 (Ambraseys and Melville, 1988; Amiran, 1951; Arieh, 1967; Ben-Menahem, 1991). Beyond the casualties, several environmental effects

10  were reported: The Jordan river flow ceased near the Damia bridge for about 21.5 hours (Willis, 1928) and a one-meter seiche wave was observed in the Dead Sea (Abel, 1927; Blankenhorn, 1927). Some evidence suggests that the earthquake was felt up to 700 kilometers from the epicenter (Ben-Menahem, 1991), although a different interpretation suggests this distance was only 300 kilometers (Ambraseys and Melville, 1988).

Compiling historical evidence, Avni (1999), in his PhD thesis, estimated the seismic intensities (MSK or Medvedev-Karkik-

15  Sponheuer scale (Medvedev et al., 1965))scale) at 133 different locations around Israel, Palestine, Jordan, Lebanon, Syria, and Egypt (Figure 4 and for locations and Ssupplementary). The curve that Avni (1999) fit to his scattered *MSK* vs *d* points, represents his basic attenuation equation, and had an $R^2$ of about 0.26. Based on the methodology proposed by Bakun and Wentworth (1997), Hough and Avni (2011) published a new attenuation equation for the Dead Sea region:Avni's (1999) basic attenuation equation yields an $R^2$ of about 0.26. Based on the methodology proposed by Bakun and Wentworth (1997),

20  Hough and Avni (2011) revised the attenuation equation for the Dead Sea region:

$$MMI(M,d) = -0.64 + 1.7M - 0.00448d - 1.67\log(d) \qquad (2)$$

where *MMI* is the Modified Mercalli Intensity (assumed to be equivalent to MSK), *M* is the magnitude and *d* is the distance from the epicenter.

Raphael and Agnon (2018) note that damage from the 1927 Jericho earthquake was higher east of the transform (on the

25  Arabian Plate) than on the west (Sinai Levant Block). This observation, consistent with their archaeoseismic findings for earthquakes in antiquity, requires further study.

**2 Methods - Multichannel Analysis of Surface Waves (MASW)**

**2.1 MASW Theory**

The MASW method is environmentally friendly, non-invasive, low-cost, rapid, robust, and provides reliable $Vs_{30}$ data (Miller et al., 2002). Multichannel records make it possible to separate different wavefields in the frequency and velocity domains. Fundamental and higher modes can be analyzed simultaneously, but generally, only the fundamental mode is used because it has the highest energy (Park et al., 1998).

The MASW method consists of three main steps: (A) Acquisition of experimental data, (B) signal processing to obtain the experimental dispersion curve, and (C) inversion to estimate $Vs_{30}$ (Figure 5). The inverse problem consists of estimating a set of parameters that describe the soil deposit, based on an experimental dispersion curve. Inversion problems based on wave propagation theory cannot be solved in a direct way due to their non-linearity. Thus, iterative methods must be used where a theoretical dispersion curve is determined for a given layer model and compared to the previously obtained experimental dispersion curve (Ryden et al., 2004). $Vs_{30}$ typically does not converge to one stable value. In other words, for the same dispersion curve, one will get slightly different $Vs_{30}$ depending on the initial model.

~~We carried out the surveys with a linear array of 24 vertical geophones (R.T. Clark's geophones with natural frequency of 4.5 Hz) at equal intervals of 2-3 meters over a total length of 46-69 meters. For the survey sound source we used a five-kilogram sledgehammer striking a twenty-centimeter square aluminum plate at variable offsets of 5, 10, 15, 20, 25 and 30 meters (both forward and reversed). The seismic data were recorded on a Geometrics Geode seismograph at a sampling rate mostly of 8 kHz for 0.5-2 seconds (Table 1). For an acceptable Signal to Noise Ratio, we used the so-called "vertical stacking" approach, which is a summation of multiple synchronized repetitions of the test (usually five times).~~

~~Rayleigh wave dispersion curves are obtained by the MASW module of the RadExPro® software, whose calculation procedure is based on a paper by Park et al. (1998), and also by the WinMASW® software. From all the dispersion images that we calculated from each offset shot (Figure 6), we choose the smoothest and clearest one (Figure 6) to compute the site's $Vs_{30}$ profile. An inversion process then finds the shear-wave velocity profile whose theoretical dispersion curve is as close as possible to the experimental curve (Figure 6). The data and coefficients are automatically inverted via genetic algorithms which represent an optimization procedure belonging to the classification of global search methods. Genetic algorithms are commonly used to generate high-quality solutions to optimization and search problems by relying on bio-inspired operators such as mutation, crossover and selection Compared to traditional linear inversion methods based on gradient methods (Jacobian matrix) these inversion techniques produce a very reliable result in terms of precision and completeness (Moro et al., 2007).~~

**2.2 Velocity model**

All models were considered to be a stack of homogeneous linear elastic layers, neglecting lateral variations in soil properties. The number of unknowns for a layered model, when considering only shear wave velocity, is three for each layer: density, thickness, and one elastic constant. Therefore, the number of unknowns is $3n-1$ (where $n$ represents the number of layers). The change in density with depth is usually small in comparison to the change in shear modulus and is normally neglected (Park et al., 1997).

**2.3 Number of layers & layer thicknesses**

The resolution of surface wave surveys decreases with depth. Thin layers are well resolved when they are close to the surface, whereas at great depth, the resolution is limited and only large changes can be detected (Foti et al., 2014). Regardless of the number of the layers of the site, $Vs_{30}$ is almost the same in each case (Figure 7). For those reasons, as well as the lack of density information, we did not restrict each model to a specific number of layers. Without boreholes and lithostratigraphic data, which is the case in our work, a useful rule of thumb is to assume layer thicknesses increasing with depth, to compensate for the decreased resolution with depth, an intrinsic shortcoming of surface wave testing (Foti et al., 2014).

**2.4 Depth of investigation**

We used a five-kilogram sledgehammer and summed up five strikes. For some sites, this type of source is insufficient to determinate a shear-wave profile down to 30 meters. In addition, at some sites, we were not able to spread the geophones at intervals of more than two meters, which limited the length of the seismic line. This length probably excludes longer wavelengths which limits the depth of investigation. Lastly, as the shear-wave velocity of the lowest frequency is higher – more data is available for deeper layers. Therefore, the penetration depth will decrease in areas with low shear wave velocity. For instance, if we can clearly detect a phase velocity of about 300 m/sec at 5 Hz, we can roughly estimate a depth of investigation of approximately 20-30 meters according to the following equation:

$$Z = \frac{\left(\dfrac{Velocity_{f_{min}}}{f_{min}}\right)}{n} ; \tag{3}$$

where $n$ equals 2-3 (Foti et al., 2014; Moro, 2015). In other words, this equation emphasizes that the depth of investigation is about a half to a third of the largest wavelength observed.

**3 Results**

We carried out the surveys with a linear array of 24 vertical geophones (R.T. Clark's geophones with a natural frequency of 4.5 Hz) at equal intervals of 2-3 meters over a total length of 46-69 meters. For the survey sound source we used a five-kilogram sledgehammer striking a twenty-centimeter square aluminum plate at variable offsets of 5, 10, 15, 20, 25 and 30 meters (both forward and reversed) (Figure 6A). The seismic data were recorded on a Geometrics Geode seismograph at a sampling rate mostly of 8 kHz for 0.5-2 seconds (Table 1). For an acceptable Signal to Noise Ratio, we used the so-called "vertical stacking" approach, which is a summation of multiple synchronized repetitions of the test (usually five times).

Rayleigh wave dispersion curves are obtained by the MASW module of the RadExPro® software, whose calculation procedure is based on a paper by Park et al. (1998), and also by the WinMASW® software. From all the dispersion images that we calculated from each offset shot (Figure 6B), we choose the smoothest and clearest one (Figure 6C) to compute the site's $Vs_{30}$ profile. An inversion process then finds the shear-wave velocity profile whose theoretical dispersion curve is as close as possible to the experimental curve (Figure 6D). The data and coefficients are automatically inverted via genetic algorithms which represent an optimization procedure belonging to the classification of global-search methods. Genetic algorithms are commonly used to generate high-quality solutions to optimization and search problems by relying on bio-inspired operators such as mutation, crossover and selection compared to traditional linear inversion methods based on gradient methods (Jacobian matrix) these inversion techniques produce a very reliable result in terms of precision and completeness (Dal Moro et al., 2007).

From 24 surveys, we succeeded in extracting $Vs_{30}$ for 19 of the 20 sites studied (the Hartuv data were too noisy for interpretation) (Table 2 and Supplementary). These would be used to recalibrate the attenuation equation arrived at by previous investigators at 133 sites19 of the 133 sites

**3.1 Velocity model**

All ground models were considered to be a stack of horizontal homogeneous elastic layers, neglecting lateral variations in soil properties. The number of unknowns for a layered model, when considering only shear-wave velocity, is three for each layer: density, thickness, and one elastic constant. Therefore, the number of unknowns is $3n$-1 (where $n$ represents the number of layers). The change in density with depth is usually small in comparison to the change in shear modulus and is normally neglected (Park et al., 1997).

**3.2 Number of layers & layer thicknesses**

The resolution of surface wave surveys decreases with depth. Thin layers are well resolved when they are close to the surface, whereas at great depth, the resolution is limited and only large changes can be detected (Foti et al., 2014). Regardless of the number of the layers at the site, $Vs_{30}$ is almost the same in each case (Figure 7). For these reasons, as well

as the lack of density information, we did not restrict each model to a specific number of layers. Without boreholes or other direct lithostratigraphic constraint, which is the case in our work, a useful rule of thumb is to assume layer thicknesses increasing with depth, to compensate for the decreased resolution with depth, an intrinsic shortcoming of surface wave testing (Foti et al., 2014).

**3.3 Depth of investigation**

We used a five-kilogram sledgehammer and summed up five strikes. For some sites, this type of source is insufficient to determinate a shear-wave profile down to 30 meters. In addition, at some sites, we were not able to spread the geophones at intervals of more than two meters, which limited the length of the seismic line. This length probably excludes longer wavelengths which limits the depth of investigation. Lastly, as the shear-wave velocity of the lowest frequency is higher - more data is available for deeper layers. Therefore, the penetration depth will decrease in areas with low shear-wave velocity. For instance, if we can clearly detect a phase velocity of about 300 m/sec at 5 Hz, we can roughly estimate a depth of investigation of approximately 20-30 meters according to the following equation:

$$Z = \frac{\frac{Velocity_{f_{min}}}{f_{min}}}{n}, \tag{3}$$

where $n$ ranges between 2 and 3 (Foti et al., 2014; Dal Moro, 2014). In other words, this equation emphasizes that the depth of investigation is about a half to a third of the largest wavelength observed.

**3.4 Recent improvement of the 1927 epicenter**

Zohar and Marco (2012) relocated the 1927 epicenter to a point near the Almog settlement. We used this most recently published epicenter to calculate new epicentral distances for the 133 sites. Since Equation 2 above is dependent upon $d$, we checked the variable scatter in the points, but found that the changes in the best-fit coefficients were very minor, so that we assumed for all purposes to use the original.

Figure 8 shows a scatter plot of the original *MMI* (assumed equivalent to MSK) vs. new $d$ for their 133 sites. Hough and Avni (2011) fit this data with a curve whose equation best describes the attenuation equation for this event. Using the mathemathical form of their curve, we calculated upper and lower limits such that 60% of the points are enclosed. This we call the 60% prediction boundary. We consider that the lithological effects probably account for much of the scatter beyond this boundry, due to amplification and de-amplification.

**4 Discussion**

5   A number of researchers have studied the 1927 event. Avni (1999) tried to reduce the impact of local geology and attempted to generate basic attenuation curves for specific azimuths. Zohar and Marco (2012) relocated the source position while Shani-Kadmie et al. (2016) studied directivity of the source pattern. None of these publications address the $Vs_{30}$ measurements. An attenuation equation with a term that depends on the $Vs_{30}$ index should lead to a better understanding of past events, and to more useful predictions of future earthquakes.

10  **4.1 Survey locations and validation**

The decision as to where exactly each survey should take place was based on Avni's thesis (Avni, 1999). Where the location was not sufficiently known, we rechecked the reference given by Avni. In most cases, there was evidence of specific damaged buildings. We tried to locate these buildings on historical maps (1927-1945). Unfortunately, most sites were located inside urban areas, where we could not carry out the seismic surveys. Therefore, we surveyed in nearby open areas as
15  close as possible to the referenced damage zones.

To validate our results, we compared them with a summary of thousands of seismic evaluations around Israel carried out over the years by the Geophysical Institute of Israel (GII), and compiled in a report by (Aksinenko and Hofstetter, 2012). These evaluations were based upon refraction and borehole velocity measurements yielding $Vs$ and/or $Vp$ values, as well as the effects of topography and geology. The spacing of their data was such that often a number of GII sites had to be averaged
20  to provide a value within several kilometers for comparison with our MASW values. However, Figure 9 shows that the GII-based values are in consistent agreement with those of the MASW. However, this comparison is a bit tricky because $Vs_{30}$ results for two sites 5 km or much less distant could be significantly different, as shown in Figure 10. Remembering that $Vs_{30}$ enters a logarithmic term, we find our approach potentially useful.

25  Table 2 lists our 24 sites alphabetically, with their respective $Vs_{30}$ values, the computed errors, and epicentral distances, $d$. The $Vs_{30}$ values vary from a low of 232 m/sec in Beit Alfa, -85 m.s.l. (Figure 11), in the thick and active alluvial plain of the famous Valley of Gilboa some 10 km from the Dead Sea rift, and a site of many millennia of agriculture. The highest value is 1,444 m/sec in Peqi'in, 680 m.s.l (Figure 11). in an area of ancient hillside orchards, and massive carbonate bedrock. On the other hand, the two Motza sites (Figure 11) lie in Emeq HaArazim (Valley of the Cedars) on the western flank of
30  Jerusalem within the massive anticlinorium of the Judean Hills, at about 570 m.s.l.. Motza 1 (1065 m/sec) is on a compacted

dirt parking lot above alluvium and the Soreq Fm., while Motza 2 (874 m/sec) is farther up the valley on a gentle hillside above the Bet Meir Fm.. Both are of similar limestone and marl composition and Cretaceous age.

**4.2 A new attenuation equation**

5  In the present case of the 1927 earthquake, the sources of the data are mostly historical documents and not strong data measurements. This makes it difficult to quantify site response into a single equation. In the practical modern attenuation relationship, $Vs_{30}$ is a crucial index. A term that depends on $Vs_{30}$ has previously been constrained for several large data sets (Abrahamson et al., 2014; Boore et al., 1997; Campbell and Bozorgnia, 2008). We chose the Boore et al. (1997) attenuation equation (Eq. (4)) in order to emphasize site response:.

$$\ln Y = b_1 + b_2(M-6) + b_3(M-6)^2 + b_5 \ln(r) + b_v \ln\left(\frac{Vs}{V_A}\right),$$ (4)

where $Y$ is the ground-motion variable (peak horizontal acceleration or pseudo-acceleration response in g), $M$ is the moment magnitude, $r$ is the epicentral distance in kilometers, $V_A$ and all $b$ terms are frequency dependent coefficients to be determined. By adding Boore et al.'s (1997) $Vs$ term to Hough and Avni (2011) attenuation equation (Eq. (2)), we suggest a new equation for the region:

$$MMI = -0.64 + 1.7M - 0.00448d - 1.67\log(d) + C_4 \ln\left(\frac{Vs_{30}}{V_A}\right),$$ (5)

where $V_A$ and $C_4$ are adjustable coefficients. The first four coefficients remain the same as we assert that the magnitude, attenuation, geometrical spreading and site response are all independent. We adopt the value of $V_A$ from Boore's (1987) equation (Eq. (4)), as it represents a single value independent of the frequency. We took formerly derived GMPE, with its coefficients, and added another term, by regressing only for the new coefficient, then optimizing $C_4$ and $V_A$ by Least Squares
20  Fitting (LSF), as shown in  Figure 12 we get the final equation:

$$MMI = -0.64 + 1.7M - 0.00448d - 1.67\log(d) - 2.1\ln\left(\frac{Vs_{30}}{655}\right),$$ (6)

**4.3 The performance of the new attenuation equation**

With these coefficients, 58% or 11 of 19 sites, were amplified or de-amplified as we expected. For the entire distance range (up to 250 km) the $Vs_{30}$ corrections leave 42% sites out of the prediction boundary (eight of nineteen sites). Seismic
25  intensities at all these eight sites are overpredicted by the attenuation equation (Eq. (2)) (Figure 13). We expect that $Vs_{30}$ at these sites will be higher than 655 m/sec in order to obtain de-amplification. However, our results show the opposite effect - these eight sites are characterized by lower $Vs_{30}$ which drives amplification. This can be caused by the fact that measurements were taken over agricultural fields, of which the upper layers (the first few meters) are characterized by low

shear-wave velocity, decreasing the average *Vs*. Another reasonable explanation is that we did not succeed in extracting the average shear-wave velocity down to 30 meters and perhaps we missed some high-velocity shear-wave layers at deeper layers. In such cases, we constrain the last layer to be thicker in order to estimate $Vs_{30}$ for all our surveys.

**5 Conclusions**

5   In this research, we investigate site amplification and de-amplification around Israel. According to previous studies (Aki, 1988; Boore, 2003; Borcherdt, 1994; Field and Jacob, 1995; Joyner and Boore, 1988), the local lithology can amplify or de-amplify wave amplitude. The commonly used modern seismic method – MASW – allowed the extraction of *Vs* profiles at 19 sites reportedly damaged by the 1927 $M_L6.2$ earthquake. We use these profiles to update the attenuation equation for the Dead Sea region by including the $Vs_{30}$ term.

10   According to this new equation, 11 sites, which constitute 58% of our measured samples, move into the 60% prediction boundary. This suggests that the prediction boundary actually encompasses over 80% of the macroseismic observations. This fit is better than any available attenuation equation for the Dead Sea region. However, as we have used only 19 sites, we should consider further research and provide wider results. Although our final equation (Eq. (6)) shows amplification and de-amplification depending on $Vs_{30}$, it does not take into consideration any other factor, such as building quality, foundation
15   depth, topography, earthquake directivity, type of fault, etc. Obviously, for better results, we must use additional methods and jointly invert some other seismic data such as: refraction (S and P waves), Horizontal to Vertical Spectral Ratio (HVSR), MASW of the transverse component of Love waves, MASW of the radial component of Rayleigh wave, Extended Spatial Auto-Correlation (ESAC), etc. Also, with these data in hand, a full inversion for the epicenter will be in order.

Despite the scarcity of data, this is the first time that an integration of historical data with shear-wave velocity profile
20   measurements improves the attenuation relation. In order to better estimate the peak ground acceleration or the seismic intensities that will be caused by future earthquakes, attenuation relations are necessary for areas characterized by high seismicity. Some of the regions of low to moderate seismicity have rich sources of historical earthquake data. The integration of historical data with modern shear-wave velocity profile measurements will lead to a better understanding of future earthquakes.

*Acknowledgments.* We thank the Neev Center for Geoinfomatics's facilities and its students. We are especially grateful to Dr. John K. Hall, founder of the Center, for his ongoing support. We are grateful to the Helmholtz Association of German Research Centers for funding this research. We thank Prof. Moshe Reshef for comments and suggestions on an earlier draft and Prof. Ran Bachrach for valuable advice. We acknowledge the contribution of Prof. Michael Weber and the geophysical deep sounding section at GFZ. Finally, we thank
30   Amit Ronen for his assistance.

**References**

Abel, F. M.: No Le Recent Tremblement de Terre en Palestine, Rev. Biblique, 36, 571–578, 1927.

Abrahamson, N. A., Silva, W. J. and Kamai, R.: Summary of the ASK14 ground motion relation for active crustal regions, Earthq. Spectra, 30(3), 1025–1055, 2014.

Aguirre, J. and Irikura, K.: Nonlinearity, liquefaction, and velocity variation of soft soil layers in Port Island, Kobe, during the Hyogo-ken Nanbu earthquake, Bull. Seismol. Soc. Am., 87(5), 1244–1258, doi:10.1144/pygs.51.3.177, 1997.

Aki, K.: Local site effects on ground motion, Earthq. Eng. Soil Dyn. II-Recent Adv. Gr. Motion Eval. Geotech. Spec. Pudlication, 20, 103–155, 1988.

Aki, K. and Richards, P. G.: Quantitative seismology, University Science Books., 2002.

Aksinenko, T. and Hofstetter, A.: 1-D semi-empricical modeling of the subsurface across Israel for site effect evaluations., 2012.

Ambraseys, N.: Earthquakes in the Mediterranean and Middle East: a multidisciplinary study of seismicity up to 1900, Cambridge University Press., 2009.

Ambraseys, N. N. and Melville, C. P.: An analysis of the eastern Mediterranean earthquake of 20 May 1202, Hist. Seism. earthquakes world, 181–200, 1988.

Amiran, D. H. .: A Revised Earthquake-Catalouge of Palastine, Isr. Explor. J., 2, 48–65, 1951.

Arieh, A.: Seismicity of Israel and Adjacent Area, Minist. Dev., 43, 1–14, 1967.

Avni, R.: The 1927 Jericho Earthquake. Comprehensive Macroseismic Analysis Based on Contemporary Sources., Ben Gurion University of the Negev, Beer Sheva (in Hebrew)., 1999.

Bakun, W. H. and Wentworth, C. M.: Estimating earthquake location and magnitude from seismic intensity data, Bull. Seismol. Soc. Am., 87(6), 1502–1521, 1997.

Ben-Menahem, A.: Four thousand years of seismicity along the Dead Sea Rift, J. Geophys. Res., 96(B12), 20195, doi:10.1029/91JB01936, 1991.

Ben-Menahem, A., Nur, A. and Moshe, V.: Tectonics, seismicity and structure of the Afro - Eurasian junction - the breaking of an incoherent plate, , 12(1), 1–50, 1976.

Blankenhorn, M.: Das Erdbeben in Juli 1927 in Palestina, Zeitschr.D.Pal, (51), 123–125, 1927.

Boore, D. M.: Simulation of ground motion using the stochastic method, Pure Appl. Geophys., 160(3), 635–676, doi:10.1007/PL00012553, 2003.

Boore, D. M., Joyner, W. B. and Fumal, T. E.: Equations for Estimating Horizontal Response Spectra and Peak Acceleration from Western North American Earthquakes: A Summary of Recent Work, Seismol. Res. Lett., 68(1), 128–153, doi:10.1785/gssrl.76.3.368, 1997.

Boore, D. M., Thompson, E. M. and Cadet, H.: Regional correlations of Vs30 and velocities averaged over depths less than and greater than 30 meters, Bull. Seismol. Soc. Am., 101(6), 3046–3059, doi:10.1785/0120110071, 2011.

Borcherdt, R., Glassmoyer, G., Andrews, M. and Cranswick, E.: Effect of site conditions on ground motion and damage, Earthq. spectra, 5(S1), 23–42, 1989.

Borcherdt, R. D.: Estimates of site-dependent response spectra for design (methodology and justification), Earthq. spectra, 10(4), 617–653, 1994.

Brawer, A. Y.: Earthquakes events in Israel from July 1927 to August 1928, 1928.

BSI: Eurocode 8 : Design of structures for earthquake resistance, 3, 2011.

Building Seismic Safety Council: NEHRP Recomendations for Seismic Regulations for New Buildings and Other Structures, Part 1 : Provisions (FEMA - 368), (Fema 368), 392, 2001.

Campbell, K. W. and Bozorgnia, Y.: NGA ground motion model for the geometric mean horizontal component of PGA, PGV, PGD and 5% damped linear elastic response spectra for periods ranging from 0.01 to 10 s, Earthq. Spectra, 24(1), 139–171, doi:10.1193/1.2857546, 2008.

Ciaccio, M. G. and Cultrera, G.: Terremoto e rischio sismico, Ediesse., 2014.

Field, E. H. and Jacob, K. H.: A comparison and test of various site-response estimation techniques, including three that are not reference-site dependent, Bull. Seismol. Soc. Am., 85(4), 1127–1143, 1995.

Foti, S., Lai, C., Rix, G. and Strobbia, C.: Surface Wave Methods for Near-Surface Site Characterization., 2014.

Guidoboni, E. and Comastri, A.: Catalogue of Earthquakes and Tsunamis in the Mediterranean Area from the 11th to the 15th Century, Istituto nazionale di geofisica e vulcanologia., 2005.

Hall, J.: The 25-m DTM (Digital Terrain Model) of Israel, Isr. J. Earth Sci., 57(3–4), 145–147, doi:10.1560/IJES.57.3-4.145, 2008.

Hall, J. F., Holmes, W. T. and Somers, P.: Northridge earthquake, January 17, 1994, Prelim. Reconnaiss. Rep., 1994.

Hough, S. E. and Avni, R.: The 1170 and 1202 CE Dead Sea Rift earthquakes and long-term magnitude distribution of the Dead Sea Fault Zone, Isr. J. Earth Sci., 58(3), 295–308, doi:10.1560/IJES.58.3-4.295, 2011.

Hough, S. E., Friberg, P. A., Busby, R., Field, E. F., Jacob, K. H. and Borcherdt, R. D.: Sediment-induced amplification and the collapse of the Nimitz Freeway, Nature, 344(6269), 853–855, doi:10.1038/344853a0, 1990.

Joyner, W. B. and Boore, D. M.: Measurement, characterization, and prediction of strong ground motion, 1988.

Kagan, E., Stein, M., Agnon, A. and Neumann, F.: Intrabasin paleoearthquake and quiescence correlation of the late Holocene Dead Sea, J. Geophys. Res. Solid Earth, 116(4), 1–27, doi:10.1029/2010JB007452, 2011.

Miller, R. D., Xia, J., Park, C. B., Survey, K. G., Hunter, J. A. and Harris, J. B.: Comparing Shear-Wave Velocity Profiles
5   Inverted From Multi- Channel Surface Wave With Borehole Measurements, , 18, 181–190, 2002.

Milne, J.: Seismology: London, Kegan Paul, Trench, Truber, 1898.

Moro, G. D.: Surface Wave Analysis for Near Surface Applications., 2015.

Moro, G. D., Pipan, M. and Gabrielli, P.: Rayleigh wave dispersion curve inversion via genetic algorithms and Marginal Posterior Probability Density estimation, , 61, 39–55, doi:10.1016/j.jappgeo.2006.04.002, 2007.

10   Park, C. B., Miller, R. D. and Xia, J.: Multi-Channel Analysis of Surface Waves ( MASW ) prepared by., 1997.

Park, C. B., Miller, R. D. and Xia, J.: Imaging dispersion curves of surface waves on multi-channel record, in SEG Technical Program Expanded Abstracts 1998, vol. 17, pp. 1377–1380, Society of Exploration Geophysicists., 1998.

Raphael, K. and Agnon, A.: Earthquakes East and West of the Dead Sea Transform in the Bronze and Iron Ages, 769–798 in Shai et al., 2018.

15   Ryden, N., Park, C. B., Ulriksen, P. and Miller, R. D.: Multimodal Approach to Seismic Pavement Testing, J. Geotech. Geoenvironmental Eng., 130(6), 636–645, doi:10.1061/(ASCE)1090-0241(2004)130:6(636), 2004.

Shani-Kadmie, S., Tsesarsky, M. and Gvirtzman, Z.: Distributed slip model for forward modeling strong Earthquakes, Bull. Seismol. Soc. Am., 106(1), 93–103, doi:10.1785/0120150102, 2016.

Shapira, A.: Redetermined magnitudes of earthquakes in the Afro-Eurasian Junction, Isr. J. Earth Sci, 28, 107–109, 1979.

20   Shapira, A., Avni, R. and Amos, N.: A new estimate for the epicenter of the Jericho earthquake of 11 July 1927, Isr. J. Earth-Sciences, 42(2), 93–96, 1993.

Singh, S. K., Lermo, J., Dominguez, T., Ordaz, M., Espinosa, J. M., Mena, E. and Quaas, R.: The Mexico earthquake of September 19, 1985-A study of amplification of seismic waves in the valley of Mexico with respect to a hill zone site, Earthq. spectra, 4(4), 653–673, 1988.

25   The Standards Institution of Israel: Design provisions for earthquake resistance of structures - SI 413, , (5), 2013.

Willis, B.: Earthquakes in the Holy Land, Bull. Seismol. Soc. Am., 18(2), 73–103, 1928.

Xia, J., Miller, R. D. and Park, C. B.: Estimation of near-surface shear-wave velocity by inversion of Rayleigh waves, Geophysics, 64(3), 691–700, doi:10.1190/1.1444578, 1999.

Zaslavsky, Y.: Questioning the applicability of soil amplification factors as defined by NEHRP (USA) in the Israel building
30   standards, Nat. Sci., 04(28), 631–639, doi:10.4236/ns.2012.428083, 2012.

Zohar, M. and Marco, S.: Re-estimating the epicenter of the 1927 Jericho earthquake using spatial distribution of intensity data, J. Appl. Geophys., 82, 19–29, doi:10.1016/j.jappgeo.2012.03.004, 2012.

[Figure]

**Figure 1: Schematic view of site amplification. Seismogram at the surface shows amplification in comparison to the seismogram located over the bedrock (modified after Ciaccio and Cultrera, 2014).**

[Figure]

Figure 2: Research area: A) Middle East area with the main tectonic elements. B) Proposed epicenters for the 1927 earthquake event with all sites that were investigated placed over a 25m DTM image (Hall, 1996). C) Detailed location of the proposed epicenters. Also shown are sites mentioned in the text: Jerusalem (J) and Nablus (N).

[Figure]

**Figure 3: Wreckage of the Winter Palace Hotel, Jericho, after the 1927 earthquake. American Colony (Jerusalem). Photo Dept., photographer.**

[Figure]

**Figure 4: Isoseismal map. The epicentral locations in red and black circles. Red and green dots are suspect amplified or de-amplified sites (respectively). Blue dots are sites which have MSK values expected from the attenuation equation (with 60% prediction boundary).**

[Figure]

**Figure 5: Multichannel Analysis of Surface Waves (MASW) technique: A. Acquisition – Using a sledgehammer as an artificial source and a linear array of geophones that receives all wavelets. B. Signal process – A fundamental mode and first higher mode over the dispersion image. C. Inversion – Final $V_s$ profile which best fits the dispersion curve.**

[Figure]

**Figure 6: Data processing (example from Binyamina): A – Raw data of four different offsets. B – The four relative dispersion images calculated from the raw data. C – Best dispersion Image (offset 15): pink dots are the analyst's dispersion curve picking. The blue line and yellow dashed line are respectively the best and the mean curves from the final model,. D – Shear-wave velocity model (Blue profile for the best one and red dashed line is the mean profile from 100 lower RMS.**

[Figure]

**Figure 7:** $V_{S30}$ as a function of a number of layers (example from Beit Alfa).

[Figure]

**Figure 8: Avni's seismic intensity (MMI) estimates of all the 133 sites. Distance is corrected according to the Zohar & Marco epicenter. Yellow dots are suspected amplified or de-amplified sites. Sites with pins are sites where we measured the *Vs* profile. Blue dots are sites which have MSK values expected from the attenuation equation (within the 60% prediction boundary).**

[Figure]

**Figure 9: Comparison between our $V_{S30}$ results (light blue) and those calculated from GII's report (red) (Aksinenko and Hofstetter, 2012).**

[Figure]

**Figure 10: Comparison between GII's closest measurements (up to 550 meters).**

[Figure]

**Figure 11: Three of the sites investigated: A) Motza 1, B) Motza 2, and C) Peqi'in. Black lines represent the seismic line location. D) The locations of the sites over a 25m DTM image (Hall, 1996). Also shown are sites mentioned in the text: Jerusalem**

[Figure]

**Figure 12: A Sensitivity analysis for calibration of the new equation.**

[Figure]

**Figure 13: Site response corrections: Yellow dots are MMI before site correction and black dots with error bars due to *Vs* uncertainty, represent the MMI after reducing site effects.**

**Acquisition parameters**

| | |
|---|---|
| **Number of geophones** | 24 |
| **Geophone spacing** | 2-3 meters |
| **Array length** | 46-69 meters |
| **Sampling rate** | 8 kHz |
| **Record length** | 0.5-2 second |
| **Receivers** | 4.5 Hz vertical |
| **Source** | 5 kg hammer |

**Table 1: Acquisition parameters.**

| ID | Site | $Vs_{30}$ [m/sec] | Error [%] | Epicentral distance |
|---|---|---|---|---|
| 1 | Acre | 261 | 13 | 131 |
| 2 | Ashkelon | 561 | 5 | 89 |
| 3 | Be'er Sheva | 359 | 8 | 91 |
| 4 | Beit Hakerem | 1436 | 12 | 29 |
| 5 | Beit Alfa | 232 | 5 | 79 |
| 6 | Binyamina | 316 | 5 | 95 |
| 7 | Givatayim | 396 | 6 | 72 |
| 8 | Herzliya | 330 | 5 | 77 |
| 9 | Jasar-Majami | 294 | 9 | 92 |
| 10 | Lod 1 | 320 | 4 | 60 |
| 11 | Lod 2 | 374 | 6 | |
| 12 | Motza 1 | 1065 | 8 | 33 |
| 13 | Motza 2 | 874 | 8 | |
| 14 | Mt. Scopus 1 | 600 | 6 | 23 |
| 15 | Mt. Scopus 2 | 582 | 5 | |
| 16 | Nahalal | 380 | 7 | 102 |
| 17 | Nahariya | 883 | 1 | 139 |
| 18 | Peqi'in | 1444 | 3 | 131 |
| 19 | Ramleh | 360 | 4 | 61 |
| 20 | Tzemach 1 | 281 | 5 | 101 |
| 21 | Tzemach 2 | 273 | 4 | |
| 22 | Tzora | 430 | 3 | 50 |
| 24 | Yavneh | 361 | 10 | 72 |

**Table 2: MASW results.**